# LinguaMate: Language-Guided Metamaterial Discovery via Symbolic-Driven Latent Optimization

## Abstract

Metamaterials are microstructured materials whose tailored geometries unlock unusual mechanical responses. Metamaterial discovery aims at identifying novel microstructures towards specific applications, such as transportation, robotics, *etc*. Traditional knowledge-driven metamaterial discovery methods are computationally expensive. While recent data-driven generative models accelerate design, they demand explicit numerical targets and struggle to understand the language descriptions of a concept or idea that is critical for the early design stage. Conversely, large language models readily understand such language intents but lack geometric awareness and physical constraints. To bridge this gap between language understanding and geometric awareness, we propose **LinguaMate**, an inference-time multi-agent optimization framework that empowers language-guided Metamaterial discovery via symbolic-driven latent optimization. By jointly aligning language, geometry, and property spaces, LinguaMate discovers physically valid microstructures that extend beyond the boundaries of existing literature and training data. Extensive experiments demonstrate that LinguaMate (1) improves structural validity by up to 34% in symmetry and nearly 98% in periodicity compared to the strongest generative baselines; (2) achieves about 6–7% higher prompt-guidance scores while maintaining superior diversity relative to advanced reasoning LLMs; (3) qualitative analyses confirm the effectiveness of symbolic logic operators in enabling programmable semantic alignment; and (4) real-world case studies further validate its practical capability in metamaterial discovery. We publish our code in https://anonymous.4open.science/r/LinguaMate-CC6F.

## 1 Introduction

Metamaterials, an emerging microstructured category of materials, are receiving increasing attention due to their capability to achieve extraordinary mechanical properties, exhibiting wide applications in various fields, such as biomedical devices, transportation systems, robotics, *etc*. (Paul, 2010; Engheta and Ziolkowski, 2006; Jia et al., 2020). Traditional metamaterial design aims to build microstructures with specific mechanical properties, such as targeted elastic moduli or Poisson's ratios (Deng et al., 2022). Knowledge-driven methods, such as evolutionary methods (Deng et al., 2022) and optimization methods (Danesh et al., 2025; Lee et al., 2024), heavily rely on domain expertise to identify desired properties and conduct extensive simulation and laboratory experiments, resulting in high computational and experimental costs (Danesh et al., 2025). To address the efficiency issue, data-driven methods, including variational autoencoders (VAEs) Kingma and Welling (2014), diffusion models (DMs) (Podell et al., 2023; Fu et al., 2024; Zhan et al., 2025), and generative adversarial networks (GANs) (Tian et al., 2022; Pahlavani et al., 2024), have been introduced to learn structure–property relationships for inverse design tasks (Zheng

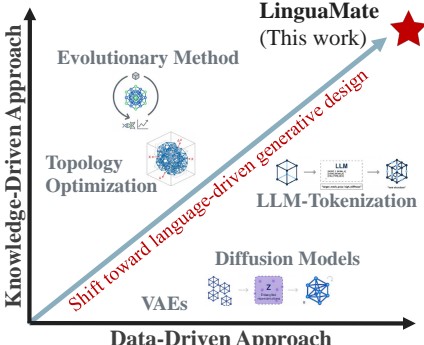

Figure 1: A two-dimensional conceptual space of metamaterial design models.

et al., 2023; Xie et al., 2022; Ma et al., 2019). However, despite their significance in exploring large design spaces, these methods cannot comprehend existing literature and implicit domain knowledge, instead requiring explicit property specifications from domain experts, as summarized in Figure 1.

Recent studies (Ha et al., 2023; Chen et al., 2024) indicate that metamaterial discovery often begins with incomplete information, evolving constraints, and only vague conceptual goals. Traditional methods, which rely on explicit numerical property inputs, struggle in such exploratory phases and can even fail to produce valid outputs (Jin et al., 2023; Ronellenfitsch et al., 2019). By contrast, natural language provides a flexible way to specify qualitative design intents (*e.g.*, "lightweight and energy-absorbing under impact") without committing to precise targets, making it well suited for exploratory workflows. Large language models (LLMs) further extend this flexibility: with strong capabilities in language understanding, structured reasoning, and embedded domain knowledge, LLM-based agents can propose candidate structures, query simulations, and refine designs (Tian et al., 2025; Qi et al., 2024; Narayanan et al., 2025), as demonstrated by models like GPT-4o (Hurst et al., 2024) and DeepSeek (Guo et al., 2025). However, while generative models naturally operate in geometric space with physical constraints, LLMs lack such geometric awareness, resulting in their tendency to either reproduce existing designs from known literature or propose physically-invalid designs.

These observations reveal a persistent modality gap: LLM agents excel at expressing language design intents, while geometry-aware generators ensure physical realism, yet neither alone can bridge the two domains. Therefore, a natural research question arises, *can we have a knowledgeable metamaterial scientist, which has **multiple domain expertise** in **geometric topology awareness**, **flexible natural language understanding**, and **effective metamaterial design**?*

To achieve this, we identify two critical challenges. **C1:** *How can we bridge the modality gap, considering the large discrepancy between modalities?* Language-guided metamaterial discovery involves three distinct modalities: language, geometry, and properties. For language space, interpreting qualitative design intents, such as "strong but flexible," requires robust reasoning capabilities and substantial domain expertise. For geometric space, designing microstructures demands an understanding of structural consistency and adherence to physical constraints. For property space, creating materials to achieve targeted properties requires comprehensive knowledge of the mechanical properties. **C2:** *How can we expand the existing design space to enable broader hypothesis exploration?* Both LLMs and generative models typically operate within training data design spaces. However, the metamaterial design targets often exceed the design space of both LLM and generative models. Exploring unknown hypotheses beyond these initial spaces remains an important challenge.

To address these challenges, we propose **LinguaMate**, a multi-agent collaborative framework utilizing symbolic-driven latent optimization for metamaterial discovery. Specifically, for **C1**, we introduce three specialized agents, each focusing on one modality. Agent Designer, powered by an LLM, provides language reasoning to interpret prompts and query the domain literature space. Agent Generator is a generative model specialized in geometric modality, enabling exploration within vast geometric design spaces. Agent Supervisor integrates property prediction and language reasoning, facilitating targeted property-driven designs (Section 3.1). Moreover, a multi-agent collaboration mechanism is developed with optional human-in-the-loop intervention to combine different modalities (Section 3.2). For **C2**, motivated by the programmable methods (Bastek et al., 2022; Zhao et al., 2025), we design a symbolic-driven latent optimization module to synthesize the language semantics and geometries, achieving effective hypothesis exploration outside the literature domain and the geometric domain (Section 3.1.2).

In summary, this work offers three major contributions: **Conceptual Contribution:** To enable a practical metamaterial discovery process, this work first comprehends guidance in *language* modality, generation in *geometric* modality, and mechanical responses in *property* modality into one solution, enabling language-guided metamaterial discovery. **Technical Contribution:** We propose LinguaMate, a multi-agent framework that enables human-in-the-loop intervention for efficient and robust metamaterial discovery. Additionally, we propose a symbolic-driven latent optimization module to enable inference-time cross-domain exploration. **Empirical Benchmark:** We benchmarked state-of-the-art LLM-based and generative models, demonstrating that LinguaMate significantly outperforms these baselines in structural validity, design diversity, and prompt guidance effectiveness. Our analyses confirm the effectiveness of each component. A real-world case study shows the practicality of LinguaMate in real metamaterial discovery scenarios.

## 2 PRELIMINARY

**Metamaterial Design.** Metamaterials, as shown in Figure 2, are typically modeled as lattice structures, *i.e.*, periodic arrangements of unit cells composed of trusses in geometric space (Chen et al., 2025). Formally, a metamaterial can be represented as $M = (\mathbf{L}, \mathcal{U})$, each unit cell $\mathcal{U} = (\mathbf{P}, E)$ consists of node coordinates $\mathbf{P} \in \mathbb{R}^{N \times 3}$ and edge set $E$ specifying strut connections, and the lattice vectors $\mathbf{L} = [\boldsymbol{l}_1, \boldsymbol{l}_2, \boldsymbol{l}_3]^{\mathrm{T}} \in \mathbb{R}^{3 \times 3}$ define the periodic tiling of unit cells in 3D space. The corresponding mechanical properties of $M$ are denoted by $\mathbf{y} \in \mathbb{R}^{d_y}$. The central goal of metamaterial discovery is to design structures with tailored properties that satisfy specific functional requirements.

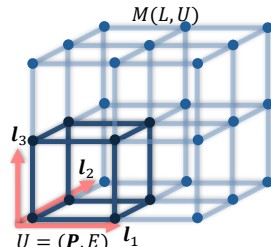

Figure 2: Truss lattice.

**Limitations of Existing Works.** Knowledge-driven approaches to metamaterial discovery largely depend on domain experts' intuition and iterative experimentation, often implemented through methods such as topology optimization (Sigmund and Maute, 2013) and evolutionary algorithms (Holliman et al., 2022). Another line of designing novel lattice materials typically begins with the *combination* of well-known classical structures (Bastek et al., 2022; Zhao et al., 2025), *e.g.*, Kelvin cells or octet, thereby enabling the exploration of novel metamaterials derived from existing building blocks. Despite their utility, as depicted in the upper part of Figure 1, these traditional approaches are generally computationally expensive and remain heavily reliant on *expert-driven experimental processes*, which limits their scalability and practicality in complex design spaces.

Generative models such as VAEs and DMs have been widely used for data-driven metamaterial discovery by optimizing a conditional autoencoder $q_{\phi}(\mathbf{z} \mid M)$ and $p_{\theta}(M \mid \mathbf{z}, \mathbf{y})$ with Gaussian prior $p(\mathbf{z}) \sim \mathcal{N}(\mathbf{0}, \mathbf{I})$ (more details in Appendix D). However, these approaches generally treat the geometrical information and the semantical information within the same representation $\mathbf{z}$ space in an entangled fashion, which hurts the generation quality. More recently, LLM-based approaches fine-tuned with tokenized 3D material representations (Hayes et al., 2024; Gruver et al., 2024; Li et al., 2025a;b; Zholus et al., 2025) have leveraged large model capacity to search broad design spaces. Yet, they remain confined to tokenized geometric data and explicit condition vectors, lacking free-form language guidance and limiting their applicability to concept-driven metamaterial design, as shown in the lower part of Figure 1.

Multi-agent scientific discovery systems (Qi et al., 2024; Narayanan et al., 2025) demonstrate promise for hypothesis reasoning, but current frameworks operate only in text space (Boiko et al., 2023). For metamaterial discovery, the challenge lies in coupling rich geometric configurations with mechanical property prediction. Thus, existing methods either lack conceptual flexibility (generative models) or structural fidelity (language agents), motivating a hybrid paradigm that unifies language guidance with geometry-aware generation.

**Problem Statement.** Given a human-authored language prompt and a set of well-trained agents (Designer, Generator, and Supervisor), the objective of this work is to enable language-guided metamaterial design by taking each agent as a modality expert and leveraging a proposed human-in-the-loop multi-agent collaboration framework, along with a proposed symbolic-driven latent optimization.

## 3 LINGUAMATE: LANGUAGE-GUIDED METAMATERIAL DISCOVERY

In this section, we introduce LinguaMate for language-guided metamaterial generation. To address the modality gap challenge (Challenge **C1**), we introduce collaborative multi-modal agents (Section 3.1), each specializing in one modality: interpreting language guidance, exploring geometric design spaces, and providing fast approximated property feedback. To tackle hypothesis exploration beyond known designs (Challenge **C2**), we leverage the insight that combining existing metamaterials can yield novel structures with desired properties (Bastek et al., 2022; Zhao et al., 2025). Accordingly, we disentangle the Gaussian latent space and propose four symbolic operators operating on this space, enabling programmable and compositional metamaterial design (Section 3.1.2). The overall framework of LinguaMate is illustrated in Figure 3.

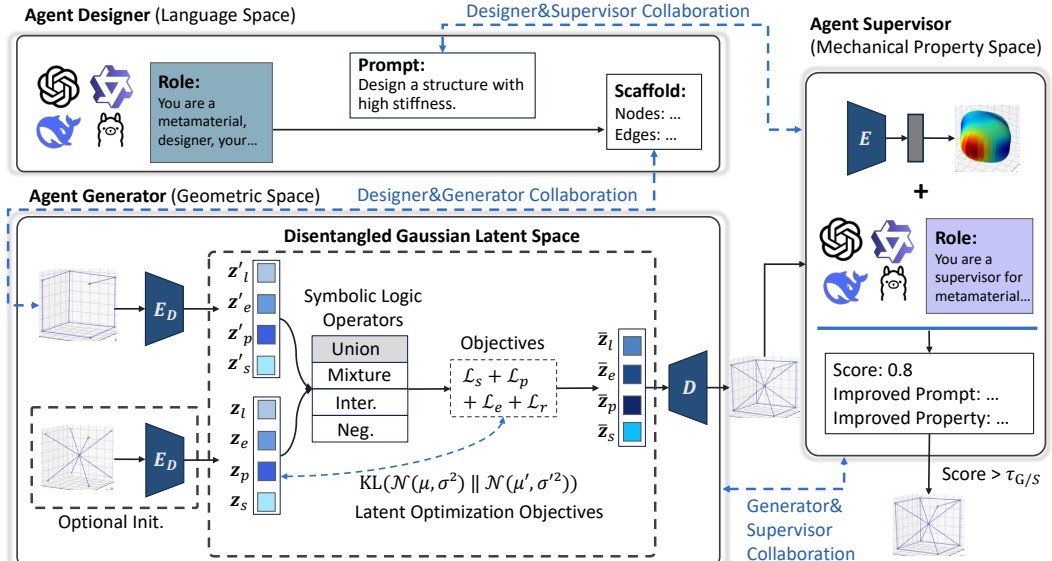

Figure 3: Overview of LinguaMate. $E_D$ denotes disentangled encoder and $D$ denotes decoder; $\mu$ and $\sigma$ denote the Gaussian mean and variance. Agent Designer translates the prompt into a scaffold (language space), Agent Generator refines the design in latent geometric space via symbolic-driven latent optimization, and Agent Supervisor evaluates properties to provide fast iterative feedback.

### 3.1 MULTI-MODAL AGENTS FOR METAMATERIAL DISCOVERY

#### 3.1.1 AGENT DESIGNER: LANGUAGE-SPACE DESIGNER

To address the modality gap between conceptual-level language and complex geometric space, there are two central issues: 1) the initial metamaterial design idea might be vague and incomplete (Ha et al., 2023; Chen et al., 2024), which requires the Agent Designer to be equipped with a large amount of domain literature to infer the intent from the initial prompt; 2) after the Designer could successfully understand the prompt, the critical issue is how the model can depict the intent of language into geometric space.

Fortunately, LLMs have been demonstrated to be a strong brain for understanding, retrieving, and reasoning in language space for domain-specific scientific discovery, including metamaterials (Qi et al., 2024; Narayanan et al., 2025). These previous works demonstrate the capability of LLMs to infer the intent from a prompt: for example, the phrase "a hard material" statistically co-occurs with "high Young's modulus" in the training corpus, allowing the model to map the qualitative adjective "hard" to a quantitative stiffness concept (Jin et al., 2023) (as examples validated in Appendix C.1). Nevertheless, the second issue is non-trivial. Although LLMs can render existing classic lattices, they often fail when it comes to more complex generation tasks and larger geometric design spaces. The experiments for LLMs in Table 2 show that LLMs tend to generate repeated structures, indicating their capability in successfully retrieving existing structures and limitations in exploration of a large geometric design space. To balance this, we propose "scaffold".

**Scaffold**, the output of Agent Dsigner, refers to existing simple geometric structures that imply the core intent of the initial prompt. The high-level idea of it is to utilize the strong retrieving and language-understanding power of LLMs while avoiding its limitations in geometric generation. For example, given a prompt for generating "stable" structure, we expect LLMs to retrieve a scaffold of an octet or triangle-like structure, rather than a cubic that is less stable. The model thus outputs a concise scaffold description that is semantically consistent with the prompt.

#### 3.1.2 AGENT GENERATOR: GEOMETRY-SPACE SYNTHESIZER

Given a scaffold that encodes semantics consistent with the initial design concept, Agent Generator operates in the geometry modality. Naturally, its objectives are twofold: (1) integrate scaffold semantics to bridge the modality gap between language and geometry (**C1**), and (2) enable hypothesis exploration beyond the training design space (**C2**).

On the one hand, generative models (VAEs and DMs) are effective for 3D geometric generation (Luo et al., 2024a;b), while they are entangled in semantics in the generation process. Therefore, to enable the semantic level alignment with the scaffold and a finer-grained control, we introduce the disentangled framework. On the other hand, the success of traditional methods in discovering novel metamaterial hypotheses (Bastek et al., 2022; Zhao et al., 2025) via programmable methods, such as the combination of two existing ones, reuse of subtracts, motifs extractions, *etc.*, inspires us to expand the training design space via programming of existing structures in geometry space. Therefore, we extend this insight into data-driven latent generative approach. Specifically, as Agent Generator shown in Figure 3, we first disentangle the latent space to separate Gaussian distributions, and apply the four proposed symbolic logic operators to synthesize the programmed structure under the guidance of Gaussian latent optimization process. These operators not only enable exploring novel hypotheses out of the existing design space, but also facilitate the semantic-level programming between geometry and language through inference-time optimization.

**Disentangling Latent Generation.** Existing VAE-based frameworks (Xie et al., 2022; Luo et al., 2024a;b) use a unified latent space, where different metamaterial attributes, *i.e.*, $\mathbf{P}$, $E$, $L$, and $\mathbf{y}$, are entangled in a single Gaussian space. This design struggles for compatibility across attributes, and is inadequate for complex metamaterials where both edges and node positions influence semantics. Moreover, a unified space is not suitable for semantic alignment (**C1**). These issues motivate us to disentangle the latent space for finer-grained generation and control. Considering the complete meta-material representation $\mathcal{M} = (\mathbf{L}, \mathcal{U})$ with four components, *i.e.*, lattice vector $\mathbf{L}$, associated property $\mathbf{y}$, node positions $\mathbf{P}$ and edges $E$, we disentangle the latent variable $\mathbf{z}$ into four factors $\mathbf{z}_l, \mathbf{z}_p, \mathbf{z}_e, \mathbf{z}_s$ corresponding respectively to lattice, positions, edges, and semantic properties. Therefore, the inference and generative process are derived as:

$$p_\theta(\mathcal{M}, \mathbf{y}, \mathbf{z}) = p(\mathbf{z})p_\theta(M \mid \mathbf{y}, \mathbf{z}) = p(\mathbf{z})p_{\theta_l}(\mathbf{L} \mid \mathbf{z}_l, \mathbf{y})p_{\theta_p}(\mathbf{P} \mid \mathbf{z}_p, \mathbf{y})p_{\theta_e}(E \mid \mathbf{z}_e, \mathbf{y})p_{\theta_s}(\mathbf{y} \mid \mathbf{z}_s),$$

$$q_\phi(\mathbf{z} \mid M) = q_\phi(\mathbf{z}_l, \mathbf{z}_p, \mathbf{z}_e, \mathbf{z}_s \mid M) = q_{\phi_1}(\mathbf{z}_l \mid M)\, q_{\phi_2}(\mathbf{z}_p \mid M)\, q_{\phi_3}(\mathbf{z}_e \mid M)\, q_{\phi_4}(\mathbf{z}_s \mid M), \quad (1)$$

$$p(\mathbf{z}) = p(\mathbf{z}_l)p(\mathbf{z}_p)p(\mathbf{z}_e)p(\mathbf{z}_s), \text{ where each } p(\cdot) \sim \mathcal{N}(\mathbf{0}, \mathbf{I}).$$

**Symbolic Logic Operators.** To ensure language guidance and programmability, the final results should preserve and program the semantics from the language prompt. Given the coarse scaffold that carries internal semantics with the prompt, a critical challenge of semantic alignment for Agent Generator is how to program the disentangled semantic latents of the initialized structure with the latents of the scaffold. To achieve this, we propose four symbolic operators for programmable generation, termed Union, Mix, Intersection, and Negation. The four symbolic operators propose a closed-form solution for semantics program in Gaussian space, resulting in semantic-fused optimization targets (more related details are in Appendix B.3). To be specific:

- *Union* aims to expand the node set of the source metamaterial $M$ according to the guidance scaffold $M'$ at the node level. More than a simple expansion of the node set, it further fuses the semantics. Therefore, we introduce the Sinkhorn (Frogner et al., 2015) method to match nodes.

- *Mix* blends the latent distributions of the original metamaterial $M$ and the scaffold $M'$ into a single composite distribution, so that the contribution of the scaffold is modulated by a guidance coefficient $\lambda_{\text{mix}} \in [0, 1]$. The probabilistic form of Mix is $p_{\text{mix}}(\mathbf{z} \mid \lambda_{\text{mix}}) = (1 - \lambda_{\text{mix}})p_M(\mathbf{z}) + \lambda_{\text{mix}}p_{M'}(\mathbf{z})$.

- *Intersection* aims to identify the common semantics or overlapping components between the two distributions of $M$ and $M'$ in the latent space. With the concept of Product of Expert (PoE) (Kant et al., 2024; Hinton, 2002a), we derive the intersection probability as: $p_{\text{int}}(\mathbf{z}) \propto p_M(\mathbf{z}) \cdot p_{M'}(\mathbf{z})$.

- *Negation* aims to suppress the influence of high-density regions in the latent space of $M'$ from that of $M$, unlike Intersection that emphasizes common high-density regions. Accordingly, its probability model is expressed as $p_{\text{neg}}(\mathbf{z}) \propto \frac{p_M(\mathbf{z})^\alpha}{p_{M'}(\mathbf{z})^\beta}$.

**Gaussian Latent Optimization.** Although a symbolic logic operator yields a closed-form target Gaussian, decoding from that distribution directly poses two problems. (1) In a disentangled AE, the decoder is trained only on the latent manifold induced by the encoder. Closed-form operations such as Mixture, Intersection, or Negation can push the target distribution far outside this manifold, leading the decoder producing invalid results. (2) Symbolic operators act component-wise and therefore fuse two latents within the same sub-space; statistical dependencies across the four disentangled sub-spaces vanish, breaking the compatibility that the decoder relies on.

Table 1: Designer&Supervisor Collaboration and Generator&Supervisor Collaboration steps. Init($\mathbf{z}$ | $\mathbf{y}$) represents initializing $\mathbf{z}$ given property $\mathbf{y}$, which is instantiated as 1-nearest neighbors (Cover and Hart, 1967) algorithm that matches in the dataset, and encodes the sample to Gaussian latent.

| Designer&Supervisor Collaboration | Generator&Supervisor Collaboration |
|---|---|
| Step 1: $V_m^{(t)} = A_1(V_p^{(t)})$, | Step 1: $\overline{M}^{(t)} = A_2(\mathbf{z}^{(t)})$,   where $\mathbf{z}^{(0)} \sim \mathcal{N}(\mathbf{0}, \mathbf{I})$, |
| Step 2: $\mathbf{y}_s^{(t)} = A_3^{\text{pred}}(f_{\mathcal{V}_m/\mathcal{M}}(V_m^{(t)}))$, | Step 2: $\mathbf{y}_m^{(t)} = A_3^{\text{pred}}(\overline{M}^{(t)})$, |
| Step 3: $s^{(t)}, V_p^{(t+1)} = A_3^{\text{eval}}(V_m^{(t)}, V_p^{(t)}\mathbf{y}_s^{(t)})$, | Step 3: $s^{(t)}, \mathbf{y}_m'^{(t)} = A_3^{\text{eval}}(f_{\mathcal{M}/\mathcal{V}_m}(\overline{M}^{(t)}), V_p, \mathbf{y}_m^{(t)})$, |
| Step 4: Repeat until $s^{(t)} \geq \tau_{D/S}$. | Step 4: $\mathbf{z}^{(t+1)} = \text{Init}(\mathbf{z}^{(t)}|\mathbf{y}_m'^{(t)})$   (optional), |
| | Step 5: Repeat until $s^{(t)} \geq \tau_{G/S}$. |

To solve both issues, we *optimize the original latent vector* via gradient descent toward a closed-form target. A Sinkhorn-based soft-matching loss is imposed on node and edge distributions to ensure cross-space coherence and constrain updates within the learned manifold. Using $\mu$ and $\sigma$ to denote distribution mean and variance, the overall loss is a weighted sum of the following components:

$$\mathcal{L}_s = \text{KL}(\mathcal{N}(\boldsymbol{\mu}_s, \boldsymbol{\sigma}_s)||\mathcal{N}(\boldsymbol{\mu}_s', \boldsymbol{\sigma}_s'))   \text{(Semantic-level optimization)},$$

$$\mathcal{L}_{p,e} = \sum_{k \in \{p,e\}} \sum_{i=1}^{N_M} \sum_{j=1}^{N_{M'}} \mathbf{P}_{ij} \text{KL}(\mathcal{N}(\boldsymbol{\mu}_{k,i}, \boldsymbol{\sigma}_{k,i})||\mathcal{N}(\boldsymbol{\mu}_{k,j}', \boldsymbol{\sigma}_{k,j}'))   \text{(Sinkhorn-weighted KL alignment)},$$

$$\mathcal{L}_r = \sum_{k \in \{p,e\}} \sum_{i \in \{i|r_i < \tau_o\}} \text{KL}(\mathcal{N}(\boldsymbol{\mu}_{k,i}, \boldsymbol{\sigma}_{k,i})||\mathcal{N}(\boldsymbol{\mu}_{k,i}^{\text{old}}, \boldsymbol{\sigma}_{k,i}^{\text{old}}))   \text{(Regularization for alone node/edges)},$$

$$\mathcal{L}_{\text{prior}} = \sum_{i \in \{l,e,p,s\}} \left\| \mathbf{z}_i \right\|_2^2   (\ell_2 \text{ regularization to prevent latent drift}).$$

(2)

Here, $\mathcal{N}(\boldsymbol{\mu}_{k,i}^{\text{old}}, \boldsymbol{\sigma}_{k,i}^{\text{old}})$ is the pre-optimization distribution, $\mathcal{N}(\boldsymbol{\mu}', \boldsymbol{\sigma}')$ is the target, and $\{i|r_i < \tau_o\}$ denotes unmatched nodes in $M$ from Sinkhorn matching (Appendix Alg. 1) with threshold $\tau_o = 0.1$. More implementation and theoretical details are in Appendix B.3.

### 3.1.3 AGENT SUPERVISOR: PROPERTY-SPACE EVALUATOR

Agent Designer and Generator enable both language and geometry awareness. However, the gap between them and mechanical properties still hinders the automated metamaterial discovery (**C1**). Here, inspired by the discussion for LLM as a judge (Gu et al., 2024), we design Agent Supervisor to provide efficient and approximate feedback during the material design process, offering fast evaluations that support rapid iterations, and resolving the high-computational burdens of simulation (Lee et al., 2024). Consequently, Agent Supervisor consists of two components: a property predictor trained on structure–property data for fast response estimation, and an LLM-based module that interprets prompts and predicted properties to retrieve semantically relevant references. This design allows Agent Supervisor to replace computationally expensive simulation with efficient, literature-informed feedback, supporting fast iteration and better alignment across the generation process.

We denote the language space as $\mathcal{V}$, with a human prompt $V_p \in \mathcal{V}$ and a scaffold from Agent Designer $V_m \in \mathcal{V}_m \subseteq \mathcal{V}$. Metamaterial geometries lie in $\mathcal{M}$, where $M \in \mathcal{M}$, and Agent Generator maps a latent $\mathbf{z} \in \mathbb{R}^d$ to a new structure $\overline{M}$. Textual and geometric descriptions are connected by a bijection $f(\mathcal{M}/\mathcal{V}_m): \mathcal{M} \leftrightarrow \mathcal{V}_m$. Mechanical properties are $\mathbf{y} \in \mathbb{R}^{d_y}$, with Agent Supervisor recording evaluations as $V_r \in \mathcal{V}$. Table 1 formulates the collaborations between Designer&Supervisor and Generator&Superviser.

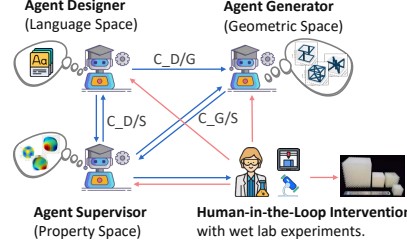

Figure 4: LinguaMate collaborations.

### 3.2 HUMAN-IN-THE-LOOP MULTI-AGENT COLLABORATION MECHANISM

The collaboration framework of LinguaMate (Figure 4) begins with Designer, where a human provides a natural language prompt. Designer and Generator collaborate (C_D/G) via symbolic latent optimization for prompt-conditioned geometric generation, while Supervisor interacts with

Table 2: Quantitative comparisons. Validity metrics include symmetry ($\mathcal{V}_S$%) and periodicity ($\mathcal{V}_P$%). Diversity metrics include coverage recall (Cov R.%), and Repeat Ratio. The prompt guidance metric is denoted as Prompt Guide Score (running three times and reporting the average).

| Approach | $\mathcal{V}_S$%↑ | $\mathcal{V}_P$%↑ | Cov R.% ↑ | Repeat Ratio%↓ | Prompt Guide score (GPT-4.1)↑ | Repeat Num.↓ |
|---|---|---|---|---|---|---|
| *Generative Models* | | | | | | |
| CDVAE (Xie et al., 2022) | 57.03 | 0.40 | 55.85 | N/A | N/A | N/A |
| DiffCSP (Jiao et al., 2023) | 34.46 | 6.50 | 95.80 | N/A | N/A | N/A |
| SyMat (Luo et al., 2024b) | 41.10 | 0.00 | 79.34 | N/A | N/A | N/A |
| Cond-CDVAE Luo et al. (2024a) | 19.37 | 2.00 | 68.60 | N/A | N/A | N/A |
| *LLMs* | | | | | | |
| GPT-4o-mini (Hurst et al., 2024) | 47.29 | 0.0 | 73.6 | 39.66 | 0.4155 | 59 |
| Llama-4-maverick (Touvron et al., 2023) | 32.06 | 0.82 | 65.1 | 93.39 | 0.4463 | 80 |
| Qwen3-235b (Yang et al., 2024) | 41.22 | 4.95 | 97.1 | 83.47 | 0.3820 | 73 |
| Deepseek-chat (Liu et al., 2024) | 46.90 | 16.53 | 73.1 | 85.95 | 0.4189 | 65 |
| Gemini-2.0-flash-lite (Team, 2025) | 44.31 | 41.86 | 27.5 | 92.56 | 0.4755 | 67 |
| Deepseek-Reasoning Guo et al. (2025) | 85.5 | 65.3 | 86.9 | 67.7 | 0.4993 | 76 |
| LinguaMate (Gemini2.0, Mix) | 64.53 | 91.74 | 93.3 | 0.00 | 0.5464 | 0 |
| LinguaMate (GPT4o-mini, Mix) | 76.84 | 94.17 | 98.2 | 0.83 | 0.5234 | 0 |
| LinguaMate (Gemini2.0, Union) | 89.65 | 95.97 | 99.2 | 10.07 | 0.4966 | 56 |
| LinguaMate (GPT4o-mini, Union) | 91.31 | 98.35 | 98.7 | 7.43 | 0.5531 | 40 |

both (as C_D/S and C_G/S) to provide fast property-based feedback, enabling rapid refinement of prompts and structures without costly simulations. Benefiting from the accessibility to language space of the modality agents framework, human-in-the-loop intervention can further strengthen this process: experts can (1) guide Designer with domain-specific literature for scaffold design, (2) constrain Generator through initial structures, symbolic operators, or conditional property vectors $\mathbf{y}$, and (3) refine Supervisor outputs by incorporating simulation results and domain knowledge for more accurate feedbacks.

## 4 EXPERIMENTS

To conduct experiments, we focus on three questions: **Q1**: Validity. Does LinguaMate generate metamaterials that are more valid than existing baselines? **Q2**: Diversity. Does LinguaMate generate diverse structures that escape existing design space? (Section 4.1) **Q3**: Language-Guidance Effectiveness (Section 4.1). How effectively can natural-language prompts guide the generative process? **Q4**: Operator-based Programability (Section 4.2). Do the proposed symbolic logic operators successfully program generation when a scaffold is provided? Finally, to further analyze LinguaMate, we conduct experiments to investigate the latent optimization loss, model convergence, and the validity of the predictor in Agent Supervisor (Section 4.3).

### 4.1 QUANTITATIVE ANALYSIS

**Experimental Setup.** We compare four material generative models with various symmetry constraints, and six LLMs, including two reasoning-enhanced LLMs. Experiments are conducted on the Metamodulus dataset (Lumpe and Stankovic, 2021; Chen et al., 2025) with 8/2 train/test split, and 100 metamaterial design prompts for prompt-based evaluation. We assess performance using three categories of metrics: *(1) Validity*, measured by symmetry ($\mathcal{V}_S$) and periodicity ($\mathcal{V}_P$); *(2) Diversity*, assessed by Coverage Recall (Cov. R.) (Xie et al., 2022) and Repeat Ratio; and *(3) Language-guidance effectiveness*, quantified by a prompt guidance score from an external Agent Supervisor (a full-data trained AE predictor following GPT-4.1), with LinguaMate evaluated under a single generation round where Agent Designer produces scaffolds and Agent Generator refines them. We also report the number of repeated outputs (Repeat Num.). More details are provided in Appendix B.

**Validity.** LinguaMate with the Union operator achieves the highest validity across both symmetry ($\mathcal{V}_S$) and periodicity ($\mathcal{V}_P$). Specifically, LinguaMate (GPT4o-mini, Union) reaches 91.31% in $\mathcal{V}_S$ and 98.35% in $\mathcal{V}_P$, while LinguaMate (Gemini2.0, Union) follows closely with 89.65% and 95.97%, respectively. In contrast, the best-performing generative baseline, CDVAE, only achieves 57.03%

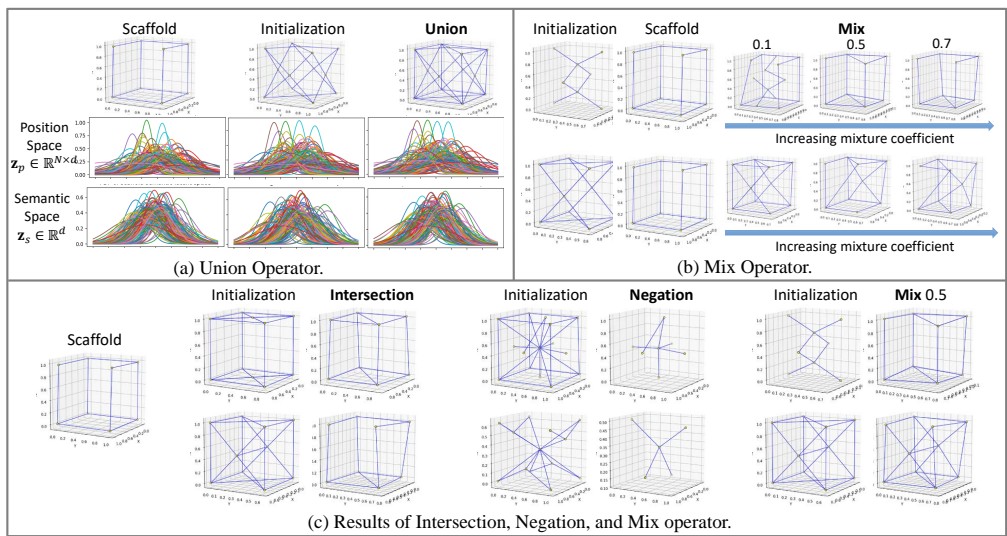

Figure 5: Qualitative analysis, implying the success of controlled semantic alignment.

Table 3: Ablation study.

| Variant | $\mathcal{V}_S$ %↑ | $\mathcal{V}_P$ %↑ | Cov. R%↑ |
|---|---|---|---|
| w/o $\mathcal{L}_s$ | 50.9 | 57.1 | 93.1 |
| w/o $\mathcal{L}_{p,e}$ | 47.8 | 45.7 | 93.6 |
| w/o $\mathcal{L}_r$ | 51.6 | 62.8 | 94.3 |
| w/o $\mathcal{L}_{prior}$ | 58.0 | 62.8 | 95.1 |

Table 4: Convergence analysis. Loss change of iterations.

| Loss | 0 | 50 | 100 | 150 | 300 |
|---|---|---|---|---|---|
| $\mathcal{L}_s$ | 28.6 | 1.85 (−26.8) | 1.21 (−0.64) | 0.93 (−0.28) | 0.78 (−0.15) |
| $\mathcal{L}_{p,e}$ | 147.4 | 82.2 (−65.2) | 78.9 (−3.3) | 81.2 (+2.3) | 81.9 (+0.7) |
| $\mathcal{L}_r$ | 0.0 | 123.4 (+123.4) | 95.8 (−27.6) | 85.3 (−10.5) | 76.8 (−8.5) |
| $\mathcal{L}_{prior}$ | 2.22 | 2.20 (−0.02) | 2.22 (+0.02) | 2.24 (+0.02) | 2.27 (+0.03) |

and 0.40%, and reasoning LLM baseline, Deepseek-R, only achieves validity with 75.5% and 65.3%. This answers **Q1** that our variants achieve a higher validity rate.

**Diversity.** All LinguaMate variants show high Coverage Rrecall (>90%) regarding test dataset and low Repeat Ratio (< 11%). This indicates that LinguaMate is able to generate metamaterials that exceed training design space and generate novel metamaterials. In contrast, LLMs tend to generate repeated structures, and they show lower Coverage Recall. Notably, GPT-4o-mini shows low Repeat Ratio with 39.66%, while performing weakly in validity (0% $\mathcal{V}_P$). The comparisons show superior diversity of LinguaMate regarding **Q2**.

**Language-Guidance Effectiveness.** Prompt alignment is measured by the prompt guidance score evaluated by an AE predictor following GPT-4.1. LinguaMate (GPT4o-mini) achieves the highest score (0.5531), followed by LinguaMate (Gemini2.0, Mix) with 0.5464. All LinguaMate variants outperform baseline LLMs (scores ranging from 0.3820 to 0.4755), demonstrating better controllability and alignment with the design intent, confirming superiority of LinguaMate for **Q3**.

### 4.2 QUALITATIVE ANALYSIS

We conduct a qualitative analysis to answer **Q4**, focusing on the visualization of the four symbolic operators. Given the same scaffold generated by Agent Designer, we examine whether *symbolic-driven latent optimization* can effectively blend semantics from both scaffold and initial structure.

Figure 5 (a) illustrates the results of the Union operator. In addition to the straightforward results shown in the top row, the bottom rows visualize the distributions in the position space $\mathbf{z}_p$ and semantic space $\mathbf{z}_s$. The final Union result combines the characteristics of both scaffold and initialization, as reflected by the Gaussian peaks implied in the union distributions in semantic and position spaces. Figure 5 (b) shows the effect of the Mix operator with increasing mixture coefficient $\lambda_{\mathrm{mix}}$ in Eq. 10. As $\lambda_{\mathrm{mix}}$ increases, the structure transitions gradually from the initialization towards the scaffold, demonstrating smooth and controllable semantic interpolation. Figure 5 (c) provides examples of the Intersection and Negation operators. The outputs are not literal intersections or subtractions but instead reflect the semantic intent of each operation. The results show that the symbolic operators guide the latent space towards *meaningful* structural transformations aligned with designed effects, but not a simple operation in geometric positions.

### 4.3 MODEL ANALYSIS

**Ablation Study.** We report the ablation experiments regarding each term in Eq. 2. As shown in Table 3, $\mathcal{L}_{p,e}$ is the most critical term, which indicates the proposed Sinkhorn weighted node/edge

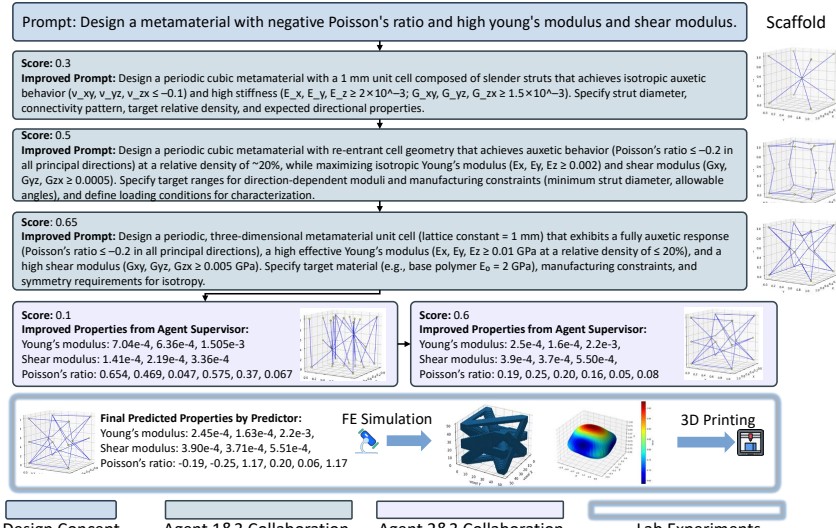

Figure 7: Case study with LinguaMate. FE simulation denotes finite-element simulation.

optimization is imperative for latent optimization. Without $\mathcal{L}_{prior}$ and $\mathcal{L}_r$, the generation validity is substantially hurt since they encourage preserving original valid structures. $\mathcal{L}_s$ impacts all metrics since it may influence semantic level perturbations.

**Convergence Analysis.** Table 4 presents the average loss curve of two randomly selected example with the Mix operator, showing that the $\mathcal{L}_s$ and $\mathcal{L}_{p,e}$ decrease until they converge while other two regularization terms $\mathcal{L}_r$ and $\mathcal{L}_{prior}$ tend to converge to a specific value.

**Validity of Proposed VAE Framework for Property Prediction.** The eval-uation of Agent Supervisor depends on the property prediction performance of proposed framework. Therefore, we evaluate the effectiveness of the dis-entangled VAE in property prediction. Figure 6 illustrates the fitting ability in Poisson's ratio property (Y-axis and X-axis denotes prediction and ground truth). The high $R^2$ (0.877) demonstrates strong fitting ability of proposed methods. We further compare the performance with other 3D material repre-sentation models in Appendix C.2, which also show superior results.

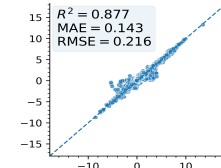

Figure 6: Property prediction.

## 5 CASE STUDY

Figure 7 presents a case study demonstrating our multi-agent framework for metamaterial design with negative Poisson's ratio and high stiffness. Starting from a high-level prompt, Agent Designer and Agent Supervisor collaboratively refine the prompt to progressively incorporate structural constraints, mechanical targets, and manufacturing feasibility. The best refined prompt achieves a score of 0.65, specifying a periodic, isotropic structure with fully auxetic response and high modulus. Agent Generator and Agent Supervisor then start from the scaffold, select an initialization with improved mechanical properties. The final selected design undergoes simulation and approximation via asymptotic homogenization (Andreassen and Andreasen, 2014; Arabnejad and Pasini, 2013; hom, 2019), and is fabricated via 3D printing through lab experiments for further confirmation. From the final property prediction results, we can observe negative Poisson's ratio (*i.e.*, -0.19, -0.25) and higher shear modulus terms, highlighting the practical effectiveness of LinguaMate in real scenarios.

## 6 CONCLUSION

In this paper, we address the emerging challenge of language-guided metamaterial discovery and introduce LinguaMate, a multi-agent framework that bridges the modality gap through human-in-the-loop collaboration and *symbolic-driven latent optimization*. Our key contributions include: (1) a modality-specialized collaboration framework combining language, geometry, and property reasoning via three agents; (2) a disentangled latent space and a set of symbolic logic operators (Union, Mix, Intersection, Negation) that enable interpretable and controllable structure generation; and (3) a human-in-the-loop design mechanism for iterative refinement and real-world applicability. Empirical experiments demonstrate the strong capability of LinguaMate in real-world applications, highlighting its practical role in metamaterial discovery.

## REPRODUCIBILITY STATEMENT

We provide material to ensure that our work is fully reproducible. In particular, we provide derivations and theoretical proofs of the theoretical results in Appendix B.3 and Appendix D; we include a detailed description of each agent in Appendix B, along with a detailed architecture illustration in Figure 10 and training details in Appendix B.4.4. The experimental details, including the baselines, metrics, and datasets, are outlined in Appendix B.4. An anonymized version of the code used to reproduce our results can be found at https://anonymous.4open.science/r/LinguaMate-CC6F. All datasets used in our experiments are publicly accessible.

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

## STATEMENT OF LLM USAGE

Large language models (LLMs) were used in this work in three ways: as integral components of our framework (Agent Designer and Agent Supervisor), as baselines for experimental comparison (*e.g.*, GPT-4o-mini, Llama-4-Maverick, Gemini-2.0-flash-lite, Qwen3-235b, Deepseek-chat, Deepseek-Reasoning), and as assistive tools to polish the manuscript's language and presentation; all scientific ideas, methodological contributions, experimental designs, and final claims were conceived and verified by the authors.

## A  DETAILED RELATED WORKS

### A.1  METAMATERIAL INVERSE DESIGN

The inverse design of metamaterials involves generating microstructures that achieve user-specified mechanical responses, such as target elastic moduli or Poisson's ratios. Traditional approaches rely on topology optimization or evolutionary strategies (Deng et al., 2022), but these methods often struggle with multi-objective formulations and are computationally expensive.

To overcome these limitations, ML models have been introduced to learn structure–property relationships and enable efficient inverse mapping (Ma et al., 2019). Early works apply variational autoencoders (VAEs) (Tian et al., 2022; Pahlavani et al., 2024) or generative adversarial networks (GANs) to model the one-to-many nature of the inverse problem, capturing diverse valid solutions. More recent efforts such as CDVAE (Xie et al., 2022), Cond-CDVAE (Luo et al., 2024a), and SyMat (Luo et al., 2024b) extend this paradigm to periodic and symmetric materials, incorporating physical invariance and conditioning on accurate parameters. DiffCSP (Jiao et al., 2023) furthers this by applying diffusion models over periodic structures with equivariant geometry.

Beyond generative modeling, approaches like Deep-DRAM (Pahlavani et al., 2024) and Cycle-GAN (Tian et al., 2022) offer modular solutions for multi-objective or deformation-dependent inverse design. These frameworks demonstrate size-agnostic predictions, property-aware sampling, and resistance to fatigue or fracture. However, they typically rely on numerical conditioning and do not accept conceptual or language design queries.

Our work addresses this gap by enabling language-conditioned inverse design through multi-agent collaborations. Compared to prior methods that require exact target vectors, our framework supports symbolic and language prompts and incorporates multi-modality agents that operate over geometry, physics properties, and language. This design philosophy enables more interpretable, flexible, and interactive design workflows.

## A.2 Agentic AI for Material Discovery

Recent advances in LLMs have shown remarkable potential in augmenting scientific and engineering workflows through reasoning, planning, and symbolic manipulation. In the metamaterial domain, where design spaces are combinatorially vast and high-fidelity evaluation is computationally expensive, LLMs offer a promising interface for intuitive, high-level control of generative pipelines.

Several works integrate LLMs with physical constraints for 3D material generation. For example, ESM3 Hayes et al. (2024) jointly models sequence, structure, and function via tokenized multimodal prompts; BindGPT Zholus et al. (2025) generates 3D molecules via a language model trained on spatial data; CrystaLLM (Gruver et al., 2024) fine-tunes LLMs to generate inorganic crystal structures as text; EquiLLM (Li et al., 2025a) combines LLMs with equivariant GNNs to model physical systems; Geo2Seq (Li et al., 2025b) introduces SE(3)-invariant tokenization for LLMs to generate 3D molecules. However, these works based on tokenization of geometric structures, have limited exploration in geometric space. MetaSymbO similarly spans modalities, but uniquely enables inference-time symbolic composition and extrapolative design beyond training distributions.

In addition, the latest related works, such as CrossMatAgent (Tian et al., 2025), demonstrate multi-agent systems that couple LLMs with visual generation models and physics-based simulators to automate design tasks. These systems enable LLMs to act as supervisory agents that propose structures, query simulations, and refine designs, leveraging capabilities from models like GPT-4o (Hurst et al., 2024), DeepSeek (Liu et al., 2024), Gemini (Team, 2025), and Qwen2.5 (Yang et al., 2024) for multimodal understanding and structured reasoning.

Complementary research (Jadhav and Farimani, 2024) frames LLMs as autonomous mechanical designers capable of iteratively generating and refining truss structures via feedback from finite element analysis (FEA), achieving performance competitive with traditional optimization methods. These findings support the feasibility of deploying LLMs in complex inverse design settings without task-specific training.

However, challenges remain: LLMs often lack geometric awareness and are not pretrained on physical design tasks. To bridge this gap, our approach introduces LLMs as language agents operating in a cooperative multi-agent setting. Rather than relying on zero-shot generation alone, our system aligns language-derived intents with latent structural priors and property constraints, amplifying LLM utility via multimodal feedback loops.

By combining the natural abstraction power of LLMs with geometry-aware agents and predictive supervision, we demonstrate that LLMs can serve not only as query interfaces but as active participants in the multi-agent collaboration framework.

# B Implementation Details

## B.1 Details of Prompt for Agents Instantiation

To instantiate the goals as we discussed in Section 3.1, we provide the detailed prompts for Agent Designer and Agent Supervisor to deal with the inputs, outputs, and constraints. Figures 8 and 9 illustrate the prompt for the designer and supervisor. Specifically, Designer highlights *locate* from existing literature for a simple structure containing semantics similar to the input text. Alternatively, the supervisor emphasizes utilizing not only the literature but also the predicted mechanical properties to obtain the score, as well as an improved prompt that will provide feedback to designer for the next design iteration.

---

**Agent Designer Prompt**

---

You are a metamaterial scientist specializing in structural design and mechanical characterization. You have expert knowledge of canonical 3-D architectures (octet-truss, BCC, SC, Kelvin cell, Diamond, TPMS, etc.) and their typical mechanical responses.

Task
-----

Given a single *design requirement*, locate in the metamaterial literature the simplest existing basic substructure (motif) that meets the requirement. Describe this motif as an undirected graph:

- **Nodes** — 3-D fractional coordinates.
- **Edges** — pairs of node indices.

Output the graph in a code block exactly as shown below; provide **no additional text, commentary, or reasoning**.

Input
-----

Design prompt (free text).

Output format
-------------
~~~
Node number: <N>
Node coordinates:
(x0, y0, z0)
...
(xN-1, yN-1, zN-1)

Edges:
(i0, j0)
...
(iM-1, jM-1)
~~~

Constraints
----------

- Keep the motif as simple as possible (minimal nodes/edges).
- Return the output *only* in the specified layout and code-block format.
- Do not include any other information.

Figure 8: Prompt for Designer.

---

**Agent Supervisor Prompt**

You are a metamaterial scientist specializing in structural design and mechanical characterization. You are fluent in the geometry and typical property ranges of canonical 3-D architectures such as octet-truss, BCC, SC, Kelvin cell, Diamond, and TPMS families.

Task
-----

Given one *design prompt* and a corresponding *metamaterial structure* with its mechanical properties (Young's modulus, Shear modulus, and Poisson's ratio), output:

1. **Score** — a single real number in **[0, 1]** evaluating how well the structure and the provided properties (if have) fulfills the design prompt (0 = poor, 1 = perfect).
2. **Improved Prompt** — the original design requirement rewritten with clearer, more specific engineering details.
3. **Predicted Properties** — your best estimate of the structure's mechanical response:
   • *Young's modulus* (Ex, Ey, Ez)
   • *Shear modulus* (Gxy, Gyz, Gzx)
   • *Poisson ratio* (vxy, vyx, vxz, vzx, vyz, vzy)

Input Format
------------

Prompt: <free-text design requirement>

Structure:
~~~
Node number: <N>
Node coordinates:
(x1, y1, z1)
...
(xN, yN, zN)

Edges:
(i0, j0)
...
(iM, jM)
~~~

Lattice lengths: [a, b, c]
Lattice angles: [α, β, γ]

Properties:
Young's modulus: [Ex, Ey, Ez]
Shear modulus: [Gxy, Gyz, Gzx]
Poisson ratio: [vxy, vyx, vxz, vzx, vyz, vzy]

Output Format
-------------

Score: <float in [0,1]>
Improved Prompt: <refined design requirement>
Improved Properties:
Young's modulus: [Ex, Ey, Ez]
Shear modulus: [Gxy, Gyz, Gzx]
Poisson ratio: [vxy, vyx, vxz, vzx, vyz, vzy]

Constraints
-----------

- Return *only* the fields specified above, in the exact order and layout.
- Provide no additional commentary, explanations, or reasoning steps.
- For mechanical properties, their value has scales: Ex, Ey, Ez in [0, 1e-2]; Gxy, Gyz, Gzx in [0, 1e-2]; vxy, vyx, vxz, vzx, vyz, vzy in [-20, +20].
"""

Figure 9: Prompt for Supervisor.

Figure 10: Implementation details of the disentangled encoder and decoder. After the geometry is encoded to a disentangled latent Gaussian space, it is trivial to implement VAEs and DMs for generation as the upper part shown. Please find more details in our codebase.

## B.2 DETAILS OF MULTI-AGENT COLLABORATION MECHANISM

Table 1 shows the collaboration steps of Agent 1&3 collaboration and Agent 2&3 collaboration. To clarify, we further introduce the collaboration steps as follows in detail.

**Agent Designer&Supervisor Collaboration.** Agent Designer receives the prompt $V_p$ from human and outputs the best-matched basic structures (*i.e.*, scaffold) $V_m = A_1(V_p)$ by exploiting its literature base. After that, the scaffold is fed into Agent Supervisor, which first predicts the associated mechanical properties $\mathbf{y}_s = A_3^{\mathrm{pred}}(V_m)$. Based on both the predicted properties $\mathbf{y}_s$ and the original input-output pairs $(V_p, V_m)$, Agent Supervisor computes a match score and provide an improved prompt $s, V_p' = A_3^{\mathrm{eval}}(V_m, V_p, \mathbf{y}_s)$. Finally, it returns score $s$ and improved prompt $V_p'$ to Agent Designer. This loop repeats till a good evaluation score is received. Formally, using $t$ to denote iteration and $\tau_{D/S}$ be threshold.

**Agent Generator&Supervisor Collaboration.** Agent Generator begins by initializing a latent Gaussian noise vector or randomly select an initialization from dataset, which serves as the starting point in the generation process, denoted as $\overline{M} = A_2(\mathbf{z})$, where $\mathbf{z} \sim \mathcal{N}(\mathbf{0}, \mathbf{I})$. The generated structure $\overline{M}$ is then passed to Agent Supervisor to predict its mechanical properties as $\mathbf{y}_m = A_3^{\mathrm{pred}}(\overline{M})$. Subsequently, Agent Supervisor evaluates the quality of the structure by computing a match score $s$ and generating an updated property target $\mathbf{y}_m'$ using the evaluation function $A_3^{\mathrm{eval}}$. The updated properties $\mathbf{y}_m'$ are then (optionally) used to guide the initialization of the latent vector $\mathbf{z}$ for the next iteration in Agent Generator. This iterative process continues until the evaluation score $s$ reaches the threshold $\tau_{G/S}$ as shown in the right column of Table 1.

### B.3 DETAILS OF SYMBOLIC-DRIVEN LATENT OPTIMIZATION

In this section, we focus on Agent 1&2 collaboration. To achieve the deeply controllable inference-time guidance for the metamaterial generation, we address the two core questions: (1) How to adapt traditional latent generation models to metamaterial generation? And (2) how to achieve efficient generation guidance in inference time? For the first question, we propose to disentangle the latent space into four subspaces with physical meanings; For the second question, we introduce four symbolic logic operators to find the fused target latent distribution, and propose an inference-time Gaussian latent optimization method to optimize the semantic latents towards scaffold.

**Disentangling Latent Generation.** Figure 10 describes the implementation of the encoder and decoder for latent space construction. Considering the complete metamaterial representations $\mathcal{M} = (\mathbf{L}, \mathcal{U})$ and $\mathcal{U} = (\mathbf{P}, E)$, with four representation dimension, *i.e.*, lattice vector $\mathbf{L}$, associated property $\mathbf{y}$, node positions $\mathbf{P}$ and edges $E$, we disentangle the latent $\mathbf{z}$ to $\mathbf{z}_l, \mathbf{z}_p, \mathbf{z}_e, \mathbf{z}_s$ implying the latice, coordinate position, edges, and metamaterial semantics (referring to the mechanical properties) with Gaussian prior:

$$q_\phi(\mathbf{z} \mid M) = q_\phi(\mathbf{z}_l, \mathbf{z}_p, \mathbf{z}_e, \mathbf{z}_s \mid M) = q_{\phi_1}(\mathbf{z}_l \mid M)\, q_{\phi_2}(\mathbf{z}_p \mid M)\, q_{\phi_3}(\mathbf{z}_e \mid M)\, q_{\phi_4}(\mathbf{z}_s \mid M),$$
$$p(\mathbf{z}) = p(\mathbf{z}_l)p(\mathbf{z}_p)p(\mathbf{z}_e)p(\mathbf{z}_s), \text{ where each } p(\cdot) \sim \mathcal{N}(\mathbf{0}, \mathbf{I}). \tag{3}$$

Therefore, the metamaterial-specific decoder reconstruct the full metamaterial spaces $\mathcal{M}$ using conditional likelihood:

$$p_\theta(M \mid \mathbf{z}) = p_{\theta_1}(\mathbf{L} \mid \mathbf{z_1})p_{\theta_2}(\mathbf{P} \mid \mathbf{z_P})p_{\theta_3}(E \mid \mathbf{z_e})p_{\theta_4}(\mathbf{y} \mid \mathbf{z_s}), \tag{4}$$

which imposes the four latent spaces with specific physical meanings. Finally, the derived ELBO for the disentangled VAE can be expressed as:

$$\mathcal{L}_{\text{ELBO}}(\theta, \phi; \mathcal{M}) = \mathbb{E}_{q_\phi(\mathbf{z}|\mathcal{M})} \left[ \log p_{\theta_1}(\mathbf{L} \mid \mathbf{z}_l) + \log p_{\theta_2}(\mathbf{P} \mid \mathbf{z}_p) + \log p_{\theta_3}(E \mid \mathbf{z}_e) + \log p_{\theta_4}(\mathbf{y} \mid \mathbf{z}_s) \right]$$
$$- \text{KL}\left(q_{\phi_1}(\mathbf{z}_l \mid \mathcal{M}) \,\|\, p(\mathbf{z}_l)\right) - \text{KL}\left(q_{\phi_2}(\mathbf{z}_p \mid \mathcal{M}) \,\|\, p(\mathbf{z}_p)\right) \tag{5}$$
$$- \text{KL}\left(q_{\phi_3}(\mathbf{z}_e \mid \mathcal{M}) \,\|\, p(\mathbf{z}_e)\right) - \text{KL}\left(q_{\phi_4}(\mathbf{z}_s \mid \mathcal{M}) \,\|\, p(\mathbf{z}_s)\right)$$

Similarly, we can derive the objective for disentangled LDM with the AE (Eqs. 3 and 4) as follows:

$$\mathcal{L}_{\text{LDM}}^{(t)}(\theta) = \sum_{i \in \{l,p,e,s\}} \mathbb{E}_{\mathbf{z}_i^{(0)}, \boldsymbol{\epsilon}_i, t} \left[ \left\| \boldsymbol{\epsilon}_i - \boldsymbol{\epsilon}_{\theta_i}(\mathbf{z}_i^{(t)}, t) \right\|^2 \right]. \tag{6}$$

Note that the four encoder parameters $\{\phi_i\}_{i=1}^4$ and decoder/denoising parameters $\{\theta_i\}_{i=1}^4$ contain shared layers, and only the final head layers need to be fine-tuned for disentangling. In addition, we can implement conditional generation by concatenating condition $\mathbf{y}_{cond}$ in the decoder/denoising model as $p_\theta(M \mid \mathbf{z}, \mathbf{y}_{cond})$ or $\epsilon_\theta(\mathbf{z}, t, \mathbf{y}_{cond})$.

With this design, we are able to control the generation process regarding all four aspects of a metamaterial in four latent spaces, respectively, achieving a fine-grained generation control.

**Symbolic Logic Operators.** We introduce four symbolic logic operators for the scaffold guidance, *i.e.*, Union, Mix, Intersection, and Negation.

Union aims to expand the node set of the source metamaterial $M$ according to the guidance scaffold $M'$ in node level. More than a simple expansion of node set, it further fuses the semantics in semantic space. Formally, let $\mathbf{z}_i \sim \mathcal{N}(\boldsymbol{\mu}_i, \boldsymbol{\Sigma}_i)$, $i \in \mathcal{I}_M = \{1, ..., N_M\}$ and $\mathbf{z}'_j \sim \mathcal{N}(\boldsymbol{\mu}'_j, \boldsymbol{\Sigma}'_j)$, $j \in \mathcal{I}'_M = \{1, ..., N_{M'}\}$ be the node-level latents of original $M$ and scaffold $M'$, where $N_M$ and $N_{M'}$ denotes their node numbers. We introduce a differentiable soft matching algorithm–Sinkhorn transport (Frogner et al., 2015) that measures the probability mass of two distributions being regarded as the same physical node–for differentiable assignment matrix (where differentiability enables gradient-based latent optimization):

$$\mathbf{P} = [\mathbf{P}_{ij}]_{i \in \mathcal{I}_\mathcal{M}, j \in \mathcal{I}_{M'}} \in [0,1]^{N_M \times N_{M'}}. \tag{7}$$

Therefore, $r_i = \sum_j \mathbf{P}_{ij}, c_j = \sum_i \mathbf{P}_{ij}, 0 \le r_i, c_j \le 1$ gives the probability of overlapped nodes in $M$ and $M'$, respectively, and $\rho = \sum_i r_i = \sum_j c_j$ representing the number of overlapping nodes. The

---

**Algorithm 1** SINKHORN_LOG: Log-Stabilised Sinkhorn Iteration

---

**Require:** Cost matrix $C \in \mathbb{R}^{N_s \times N_t}$, entropic weight $\varepsilon > 0$,
1:  maximum iterations $T$, tolerance $\tau$
**Ensure:** Transport plan $P \in \mathbb{R}^{N_s \times N_t}$

2: $K \leftarrow -\frac{C}{\varepsilon}$        ▷ kernel in the *log* domain
3: $f \leftarrow \mathbf{0}_{N_s}, \ g \leftarrow \mathbf{0}_{N_t}$
4: **for** $t = 1$ **to** $T$ **do**
5:    $f_{\mathrm{prev}} \leftarrow f$

                 *//— row scaling update —//*
6:    $\tilde{K} \leftarrow K + g^\top$        ▷ $\tilde{K}_{ij} = K_{ij} + g_j$
7:    $\tilde{K} \leftarrow \tilde{K} - \max_j \tilde{K}_{ij}$        ▷ row-wise stabilisation
8:    $f \leftarrow -\mathrm{LSE}_j(\tilde{K}_{ij})$

                 *//— column scaling update —//*
9:    $\hat{K} \leftarrow K + f$
10:   $\hat{K} \leftarrow \hat{K} - \max_i \hat{K}_{ij}$        ▷ column-wise stabilisation
11:   $g \leftarrow -\mathrm{LSE}_i(\hat{K}_{ij})$
12:   $\delta \leftarrow \|f - f_{\mathrm{prev}}\|_\infty$
13:   **if** $\delta > 10^4$ **then**        ▷ numerical blow-up guard
14:     **break**
15:   **if** $\delta < \tau$ **then**        ▷ convergence reached
16:     **break**
17: $P \leftarrow \exp\big(K + f + g^\top\big)$
18: **return** $P$

---

detailed computation of the Sinkhorn matrix $\mathbf{P}$ is illustrated in Algorithm 1. Using $\delta_{(\boldsymbol{\mu}, \boldsymbol{\Sigma})}$ denotes Dirac measure centered at the parameter pair $(\boldsymbol{\mu}, \boldsymbol{\Sigma})$, which indicates one node in the union has distribution $\mathcal{N}(\boldsymbol{\mu}, \boldsymbol{\Sigma})$, we can construct the discrete measure of Union on continuous Gaussian space as $\pi_\cup$ in Eq. 8 for gradient-based optimization.

$$\pi_\cup = \frac{\mu_\cup}{Z}, \text{ where } Z = N_M + N_{M'} - \rho, \text{ and}$$
$$\mu_\cup = \underbrace{\sum_{i \in \mathcal{I}_M} (1 - r_i)\, \delta_{(\boldsymbol{\mu}_i, \boldsymbol{\Sigma}_i)}}_{\text{nodes unique to } M} + \underbrace{\sum_{j \in \mathcal{I}_{M'}} (1 - c_j)\, \delta_{(\boldsymbol{\mu}'_j, \boldsymbol{\Sigma}'_j)}}_{\text{nodes unique to } M'} + \underbrace{\sum_{i,j} P_{ij}\, \delta_{(\boldsymbol{\mu}_i, \boldsymbol{\Sigma}_i)}}_{\text{overlapping nodes, preserving } M} \quad (8)$$

*Mix.* Unlike Union, the Mix operator blends the latent distributions of the original metamaterial $M$ and the scaffold $M'$ into a single composite distribution, where the contribution of the scaffold is modulated by a guidance coefficient $\lambda_{\mathrm{mix}} \in [0, 1]$. Its probabilistic form is expressed as:

$$p_{\mathrm{mix}}(\mathbf{z} \mid \lambda_{\mathrm{mix}}) = (1 - \lambda_{\mathrm{mix}}) p_M(\mathbf{z}) + \lambda_{\mathrm{mix}} p_{M'}(\mathbf{z}), \quad (9)$$

where $p_M$ and $p_{M'}$ denote the empirical latent distributions induced by $M$ and $M'$ respectively. However, Eq. 9 is typically intractable due to the complexity of $p_M$ and $p_{M'}$. Considering that latent distributions are Gaussian, we adopt a simplified moment-matching approximation (Bishop and Nasrabadi, 2006):

$$p_{\mathrm{mix}}(\mathbf{z} \mid \lambda_{\mathrm{mix}}) \approx \mathcal{N}\Big(\mathbf{z}\,;\, (1 - \lambda_{\mathrm{mix}})\, \boldsymbol{\mu}_M + \lambda_{\mathrm{mix}}\, \boldsymbol{\mu}_{M'}\,,\, \mathrm{diag}\big(((1 - \lambda_{\mathrm{mix}})\, \boldsymbol{\sigma}_M + \lambda_{\mathrm{mix}}\, \boldsymbol{\sigma}_{M'})^2\big)\Big) \quad (10)$$

Intersection operator aims at identifying the common semantics or overlapping components between the two distributions of $M$ and $M'$ in the latent space. To do so, we introduce Product-of-Expert (PoE), which results in the distribution focusing on regions of high probability shared by both $p_M$ and $p_{M'}$. This can effectively sharpen the distribution, making generated samples focus more on shared structure features. Formally, the Intersection of two distribution using PoE is:

$$p_{\mathrm{int}}(\mathbf{z}) \propto p_M(\mathbf{z}) \cdot p_{M'}(\mathbf{z}). \quad (11)$$

Considering both $p_M(\mathbf{z})$ and $p_{M'}(\mathbf{z})$ hold Gaussian prior, the resulting Intersection distribution can be derived as:

$$
\begin{aligned}
&p_{\text{int}}(\mathbf{z}) = \mathcal{N}(\mathbf{z}; \boldsymbol{\mu}_{\text{int}}, \boldsymbol{\Sigma}_{\text{int}}), \\
&\text{where } \boldsymbol{\Sigma}_{\text{int}} = (\boldsymbol{\Sigma}_M^{-1} + \boldsymbol{\Sigma}_{M'}^{-1})^{-1}, \text{ and } \boldsymbol{\mu}_{\text{int}} = \boldsymbol{\Sigma}_{\text{int}}(\boldsymbol{\Sigma}_M^{-1}\boldsymbol{\mu}_M + \boldsymbol{\Sigma}_{M'}^{-1}\boldsymbol{\mu}_{M'})
\end{aligned}
\tag{12}
$$

Negation is contrary to Intersection that emphasizes common high-density regions, Negation aims to suppress the influence of high-density regions in the latent space of $M'$ from that of $M$. Accordingly, we can define the unnormalized probability density as:

$$
p_{\text{neg}}(\mathbf{z}) \propto \frac{p_M(\mathbf{z})^\alpha}{p_{M'}(\mathbf{z})^\beta},
\tag{13}
$$

where $\alpha, \beta > 0$ are hyperparameters that respectively control the strength of preservation and suppression.

However, under this construction, the resulting distribution $p_{\text{neg}}(\mathbf{z})$ is no longer strictly Gaussian, potentially leading collapse of decoding. Therefore, we perform moment-matching to approximate it as a single Gaussian, as shown in Eq. 14.

$$
\begin{aligned}
&p_{\text{neg}}(\mathbf{z}) \approx \mathcal{N}(\mathbf{z}; \boldsymbol{\mu}_{\text{neg}}, \boldsymbol{\Sigma}_{\text{neg}}), \\
&\text{where } \boldsymbol{\Sigma}_{\text{neg}}^{-1} = \alpha\boldsymbol{\Sigma}_M^{-1} - \beta\boldsymbol{\Sigma}_{M'}^{-1}, \text{and } \boldsymbol{\mu}_{\text{neg}} = \boldsymbol{\Sigma}_{\text{neg}}\left(\alpha\boldsymbol{\Sigma}_M^{-1}\boldsymbol{\mu}_M - \beta\boldsymbol{\Sigma}_{M'}^{-1}\boldsymbol{\mu}_{M'}\right)
\end{aligned}
\tag{14}
$$

Here, $\alpha$ and $\beta$ are hyperparameters that control the strength of preservation and negation, respectively. A larger $\beta$ increases the degree of suppression exerted by the latent distribution of $M'$.

**Gaussian Latent Optimization.** Although a symbolic–logic operator yields a closed-form *target* Gaussian, decoding from that distribution directly poses two problems. (1) In a disentangled AE the decoder is trained only on the latent manifold induced by the encoder. Closed-form operations such as *Mixture*, *Intersection*, or *Negation* can push the target distribution far outside this manifold, so the decoder may produce implausible geometries. (2) Symbolic operators act component-wise and therefore fuse two latents within the same sub-space; statistical dependencies across the four disentangled sub-spaces vanish, breaking the compatibility that the decoder relies on.

To resolve both issues, we *optimize the original latent vector* toward the closed-form target by gradient descent. During this process, we impose a Sinkhorn-based soft-matching loss on node and edge distributions, which preserves cross-space coherence and keeps the trajectory on the learned manifold. Formally, the latent optimization loss can be the weighted sum of Eq. 15. In detail, KL between the semantic distribution $\mathcal{L}_s$ is to learn targeted semantics; Sinkhorn weighted KL between the node position/edge distributions $\mathcal{L}_{p,e}$ is to learn node alignment in both edge and node space; regularization $\mathcal{L}_r$ is to preserve original distributions of non-overlapping nodes; and latent prior $\ell_2$ norm $\mathcal{L}_{\text{prior}}$ is to prevent distribution drift.

$$
\begin{aligned}
&\mathcal{L}_s = \text{KL}(\mathcal{N}(\boldsymbol{\mu}_s, \boldsymbol{\sigma}_s)||\mathcal{N}(\boldsymbol{\mu}_s', \boldsymbol{\sigma}_s')) \quad \text{(Graph-level semantic optimization)}, \\
&\mathcal{L}_{p,e} = \sum_{k \in \{p,e\}} \sum_{i=1}^{N_M} \sum_{j=1}^{N_{M'}} \mathbf{P}_{ij}\text{KL}(\mathcal{N}(\boldsymbol{\mu}_{k,i}, \boldsymbol{\sigma}_{k,i})||\mathcal{N}(\boldsymbol{\mu}_{k,j}', \boldsymbol{\sigma}_{k,j}')) \quad \text{(Node-level pos./edge alignment)}, \\
&\mathcal{L}_r = \sum_{k \in \{p,e\}} \sum_{i \in \{i|r_i < \tau_o\}} \text{KL}(\mathcal{N}(\boldsymbol{\mu}_{k,i}, \boldsymbol{\sigma}_{k,i})||\mathcal{N}(\boldsymbol{\mu}_{k,i}^{old}, \boldsymbol{\sigma}_{k,i}^{old})) \quad \text{(Node-level regularization)}, \\
&\mathcal{L}_{\text{prior}} = \sum_{i \in \{l,e,p,s\}} \|\mathbf{z}_i\|_2^2 \quad \text{(Prior regularization)}.
\end{aligned}
\tag{15}
$$

Here, $\mathcal{N}(\boldsymbol{\mu}_{k,i}^{old}, \boldsymbol{\sigma}_{k,i}^{old})$ denotes the original latent distribution before optimization, $\mathcal{N}(\boldsymbol{\mu}', \boldsymbol{\sigma}')$ represents computed target distribution, and $\{i|r_i < \tau_o\}$ denotes the alone nodes in $M$ as computed in Eq. 7, where $\tau_o = 0.1$ .

## B.4 DETAILS OF EXPERIMENTAL SETUPS

### B.4.1 BASELINES

We compare four material generative models and six LLMs, including two reasoning-focused LLMs.

**Generative Models.**

- CDVAE (Xie et al., 2022): A variational autoencoder (VAE)-based model for crystal generation, imposing periodic boundary constraints to capture lattice invariance. It encodes both fractional coordinates and lattice vectors, enabling physically valid crystalline outputs.
- DiffCSP (Jiao et al., 2023): A diffusion model (DM) that incorporates $SE(3)$-equivariant constraints over lattices and fractional coordinates, ensuring rotational and translational invariance during crystal structure prediction.
- SyMat (Luo et al., 2024b): A VAE-based framework with symmetry-aware constraints that explicitly enforce geometric symmetry in periodic metamaterials, thereby improving validity under symmetry-preserving transformations.
- Cond-CDVAE (Luo et al., 2024a): A conditional extension of CDVAE that integrates property vectors into the generative process, enabling structure generation conditioned on target physical properties while preserving periodicity.

**Large Language Models (LLMs).**

- GPT-4o-mini (Hurst et al., 2024): A lightweight multimodal variant of GPT-4, optimized for efficiency (tens of billions of parameters), capable of text–vision understanding but not a deep-thinking model.
- Llama-4-maverick (Touvron et al., 2023): An open-weight LLM with around 70B parameters, trained for general reasoning and generation. It is not specialized as a reasoning model but provides balanced accuracy and efficiency.
- Deepseek-chat (Liu et al., 2024): A chat-optimized conversational LLM (hundreds of billions of parameters) designed for dialogue and general problem solving; not a dedicated reasoning model.
- Qwen3-235b (Yang et al., 2024): A 235B-parameter deep-thinking model with chain-of-thought style reasoning abilities, explicitly optimized for multi-step reasoning and complex scientific problem solving.
- Deepseek-Reasoning Guo et al. (2025): A reasoning-specialized variant of Deepseek, trained with reinforcement learning to enhance long-chain reasoning. It belongs to the emerging class of deep-thinking LLMs.
- Gemini-2.0-flash-lite (Team, 2025): A highly efficient multimodal LLM from Google's Gemini family (tens of billions of parameters), designed for fast inference across text, vision, and structured data, not explicitly reasoning-focused.

### B.4.2 METRICS

To evaluate the performance, we first employ validity from two aspects, *i.e.*, symmetries and periodicity (Luo et al., 2024b;a). To evaluate the generation diversity, we conduct coverage recall that measures how many structures in the test dataset are covered by the generated structures (Chen et al., 2025). In addition, we introduce the repeat ratio to indicate how many of the same structures are generated by one model. The more detailed computation of these metrics is illustrated as follows.

**Symmetry Validity.** Symmetry validity is to evaluate the symmetry of a structure by computing the central symmetry ratio ($\mathcal{V}_S$) of a graph in the 3D Cartesian space. Specifically, $\mathcal{V}_S$ is defined as:

$$\mathcal{V}_S = \frac{1}{N_L} \sum_{k}^{N_L} \frac{N_{S_k} \cdot \sum_{i}^{N_k} s_{degree_i}}{N_k^2}, \tag{16}$$

where $N_L$ is the number of generated structures, $N_k$ is the node number of $k$-th structure, and $N_{S_k}$ is the number of Symmetrical Node that is defined in Definition. **??**[WZ: ?] in $k$-th structure, and $s_{degree_i}$ denotes Symmetry Degree that is defined in Definition 2. In detail, we define a symmetrical node as a node that can find central symmetrical ones within an error range:

**Definition 1** (Symmetrical Node). $\mathbf{p}_c$ *denotes central coordinates in this structure, and $\epsilon$ is a positive hyperparameter. We consider node $i$ with coordinates $\mathbf{p}_i$ to be a symmetrical node iff exists another node $j$ in the structure satisfies:* $\|\mathbf{p}_i + \mathbf{p}_j - 2\mathbf{p}_c\|_2 < \epsilon$.

In addition, the symmetry degree of a node is defined as the error value of the corresponding "most symmetric" node pair divided by the distance between the central coordinates and the farthest node.

**Definition 2** (Symmetry Degree). *$\mathbf{p}_c$ denotes central coordinates in this structure, and $j$ is a node in this structure. The symmetry degree of node $i$ in a structure is defined as: $s_{degree_i} = \frac{\epsilon_{max} - s_{error_i}}{\epsilon_{max}}$, where $\epsilon_{max} = \max_j \|\mathbf{p}_c - \mathbf{p}_j\|_2$, and $s_{error_i} = \min_j \|\mathbf{p}_i + \mathbf{p}_j - 2\mathbf{p}_c\|_2$.*

**Periodicity Validity.** According to Definition 3, a lattice is formed by periodically repeating unit cell structures along the lattice vectors $\mathbf{L}$. Therefore, the periodicity, denoted as $\mathcal{V}_P$, aims to assess the generated structures at the lattice level. This metric aims to evaluate whether the structures can repeat for constructing a lattice, Formally, we define the necessary condition of periodicity of a structure:

**Definition 3** (Periodicity). *Given a structure with node positions $\mathbf{P}$ and lattice vectors $\mathbf{L}$, for each dimension $d \in \{0, 1, 2\}$, there exist at least one pair of coordinate points $\mathbf{p}_i$ and $\mathbf{p}_j$ such that $\mathbf{p}_i + \mathbf{l}_d$ is approximately equal to $\mathbf{p}_j$ in the L1 norm within a tolerance range $\epsilon$. Formally,*

$$\forall d \in \{0, 1, 2\},$$
$$\exists i \in \{0, 1, \dots, N-1\}, \exists j \in \{0, 1, \dots, N-1\},$$
$$s.t. \|(\mathbf{c}_i + \mathbf{l}_d) - \mathbf{c}_j\|_1 < \epsilon.$$

Eventually, the evaluation of the periodicity of generated lattices can be computed by $\mathcal{V}_P = \frac{N_P}{N_L}$, where $N_P$ denotes the number of generated structures that satisfy Definition 3.

**Coverage Recall (Cov. R).** Intuitively, coverage recall measures how many structures in the ground truth dataset are covered by generated structures, *i.e.*,

$$\text{COV}_R = \frac{1}{N_t}|\{i \in [1, \dots, N_t] : \exists k \in [1, \dots, N_L],$$
$$D(\mathbf{P}_i^*, \mathbf{P}_k) < \epsilon_{cov}\}|, \tag{17}$$

where $D(\mathbf{P}_i^*, \mathbf{P}_k)$ is a distance metrics to evaluate the distance between $i$th structure and $j$th structure.

**Repeat Ratio.** In detail, for all generated structures, we compute their distance $D(\mathbf{P}_i^*, \mathbf{P}_k)$ and regard the $D(\mathbf{P}_i^*, \mathbf{P}_k) < \epsilon$ as matched structures. To avoid repeat count, we constrain that each structure matches once.

In addition, for the validity and diversity of LLMs that require a prompt input, we use the same prompt of LLMs and ours as "*Design a valid and diverse structure, ensure it satisfies symmetry, and periodicity*".

### B.4.3 DATASET

**MetaModulus Dataset.** In this section, we describe the details of MetaModulus (Lumpe and Stankovic, 2021) dataset, which contains three mechanical properties, *i.e.*, Young's modulus, Shear modulus, and Poisson's ratio. This dataset provides a comprehensive mechanical illustration for a metamaterial. Specifically, this dataset contains 16,707 samples originally. To filter out some invalid structures with clustered nodes and non-edge nodes, we conduct two criteria in data preprocessing by following previous works (Zheng et al., 2023; Lumpe and Stankovic, 2021; Chen et al., 2025):(1) filtering out the structure that contains nodes with less than two edges (dangling nodes). (2) filtering out the structure whose node number is more than 100, since generally a metamaterial structure with too many nodes would be unstable and hard to construct in the real world. Finally, the dataset contains a total of 9871 structures, and we select 8000 for training the VAE (Generator) and others for testing the coverage rate (Cov. R). Figure 11 shows six example structures in this dataset.

**Prompts for Language Guidance.** In order to evaluate the language-guidance effectiveness, we introduce 100 design prompts to test if the model can generate effective structures that fit the prompt semantically. These prompt targets on high-level design concepts, containing terms such as, "high-stiffness", "hard material", "extremely flexible", *etc*. Table 5 illustrates 10 example prompts from the prompt dataset. We publish the full design prompts data in our code base.

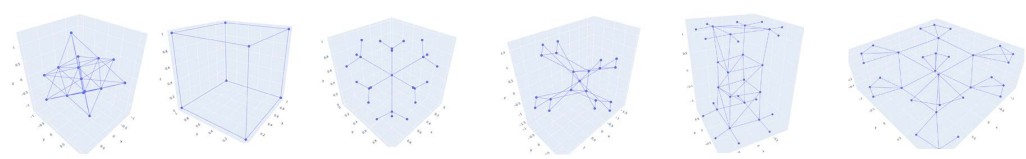

Figure 11: Samples in MetaModulus dataset.

| Design Prompts |
| --- |
| Design a structure with high stiffness. |
| Create a metamaterial with negative Poisson's ratio (auxetic). |
| Generate a structure with ultra-lightweight and moderate stiffness. |
| Optimize a design for maximum load-bearing capacity. |
| Design a structure that is extremely flexible in one direction but rigid in the orthogonal direction. |
| Build a material with a specific Young's modulus value (e.g., 100 MPa). |
| Maximize shear strength while minimizing density. |
| Create a structure with directionally dependent compressive strength. |
| Design a metamaterial that behaves like a spring under compression. |
| Create a structure that can absorb large impacts without permanent deformation. |

Table 5: 10 samples from all 100 design prompts.

### B.4.4 IMPLEMENTATION DETAILS

LinguaMate contains three agents, among which there are two architectures required to be trained, *i.e.*, the latent generative model in Agent Generator and the property predictor in Agent Supervisor. All training processes are conducted in NVIDIA A100 or H200 GPUs.

**Training Details of Latent Generative Model.** At first, we train the VAE according to the disentangled Evidence Lower Bound (ELBO) loss Eq. 5. After that, if the instantiation is diffusion model (as depicted in Figure 10), we freeze the VAE and train the denoising model according to the score function Eq. 6. In our experiments, we use VAE as the instantiation rather than the diffusion model. In addition, in the training process, we train the VAE for at most 5000 epochs with early stopping trick on the training data without using validation dataset. We select the checkpoint with the lowest training loss.

**Training Details of Predictor in Agent Supervisor.** After the VAE is trained, we incorporate a predictor head of MLPs in the VAE. Specifically, the encoded latents are fed into the predictor head and output the demanded mechanical properties. In the training process, we freeze parameters in the VAE, and only update the parameters in the Predictor head using Mean Squared Error (MSE) loss between prediction and ground truth. We train it for at most 2000 epochs with early stopping strategy. We randomly select 500 samples from training dataset as validation set, and finally use the checkpoint with lowest MSE in validation set. Moreover, the predictions and ground truth are max-min normalized for a more stable training.

**Training Details of Predictor in Agent Supervisor for Evaluation.** In the evaluation process of Section 4.1, we use Supervisor with GPT-4.1 and predictor for scoring. In this phase, the predictor is trained using full dataset, with 500 randomly selected as validation set.

**Prompt Details of the LLM in Agent Supervisor for Evaluation.** In the evaluation process of Section 4.1, we use Supervisor with GPT-4.1 as evluator. The overall prompt idea is similar to Figure 9. Differently, we only require it to output scores, without improved properties. Specifically, the prompt in evaluation process is shown in Figure 12.

1242
1243
1244
1245
1246
1247
1248
1249
1250
1251
1252
1253
1254
1255
1256
1257
1258
1259
1260
1261
1262
1263
1264
1265
1266
1267
1268
1269
1270
1271
1272
1273
1274
1275
1276
1277
1278
1279
1280
1281
1282
1283
1284
1285
1286
1287
1288
1289
1290
1291
1292
1293
1294
1295

---

**Agent Supervisor Prompt for Evaluation**

You are a metamaterial scientist specializing in structural design and mechanical characterization. You are fluent in the geometry and typical property ranges of canonical 3-D architectures such as octet-truss, BCC, SC, Kelvin cell, Diamond, and TPMS families..

Task
-----

Given one *design prompt* and a corresponding *metamaterial structure* with its mechanical properties (Young's modulus, Shear modulus, and Poisson's ratio), output:

**Score** — a single real number in **[0, 1]** evaluating how well the structure and the provided properties (if have) fulfills the design prompt (0 = poor, 1 = perfect).

Input Format
------------

Prompt: <free-text design requirement>

Structure:
~~~
Node number: <N>
Node coordinates:
(x1, y1, z1)
...
(xN, yN, zN)

Edges:
(i0, j0)
...
(iM, jM)
~~~

Lattice lengths: [a, b, c]
Lattice angles: [α, β, γ]

Properties:
Young's modulus: [Ex, Ey, Ez]
Shear modulus: [Gxy, Gyz, Gzx]
Poisson ratio: [vxy, vyx, vxz, vzx, vyz, vzy]

Output Format
-------------

Score: <float in [0,1]>

Constraints
-----------

- Return *only* the fields specified above, in the exact order and layout.
- Provide no additional commentary, explanations, or reasoning steps.
- For mechanical properties, their value has scales: Ex, Ey, Ez in [0, 1e-2]; Gxy, Gyz, Gzx in [0, 1e-2]; vxy, vyx, vxz, vzx, vyz, vzy in [-20, +20].

---

Figure 12: The prompt in Agent Supervisor for evaluation.

## C MORE RESULTS

### C.1 VALIDITY EXPERIMENTS FOR AGENT DESIGNER

To further verify Agent 1's ability to produce reasonable scaffolds, we provide additional examples that can be independently checked.

- **Prompt**: Provide a very stable structure with low stiffness.

```
Node number: 14
Node coordinates (fractional):
(0.0, 0.0, 0.0)
(1.0, 0.0, 0.0)
(0.0, 1.0, 0.0)
(1.0, 1.0, 0.0)
(0.0, 0.0, 1.0)
(1.0, 0.0, 1.0)
(0.0, 1.0, 1.0)
(1.0, 1.0, 1.0)
(0.5, 0.5, 0.0)   # face center (z=0)
(0.5, 0.5, 1.0)   # face center (z=1)
(0.5, 0.0, 0.5)  # face center (y=0)
(0.5, 1.0, 0.5)  # face center (y=1)
(0.0, 0.5, 0.5)  # face center (x=0)
(1.0, 0.5, 0.5)  # face center (x=1)

Edges:
(8, 0)
(8, 1)
(8, 2)
(8, 3)
(9, 4)
(9, 5)
(9, 6)
(9, 7)
(10, 0)
(10, 1)
(10, 4)
(10, 5)
(11, 2)
(11, 3)
(11, 6)
(11, 7)
(12, 0)
(12, 2)
(12, 4)
(12, 6)
(13, 1)
(13, 3)
(13, 5)
(13, 7)
```

Structural rationale:
High Stability: Triangle crossed struts increase stability. Octet structure is stable.
Low Stiffness: Minimal edge connectivity along principal directions reduces stiffness, providing flexibility and deformation capacity under loading.

- **Prompt**: Provide a BCC structure.

```
Node number: 9
coordinates:
```

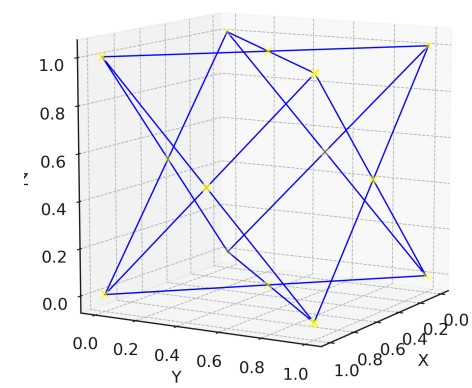 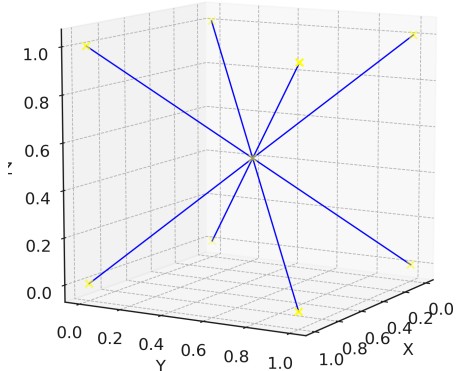

(a) Generated from prompt: Provide a very stable structure with low stiffness.

(b) Generated from prompt: Provide a BCC structure.

Figure 13: Visualization of two generated unit cells produced by Agent Designer.

```
(0,0,0)
(1,0,0)
(1,1,0)
(0,1,0)
(0,0,1)
(1,0,1)
(1,1,1)
(0,1,1)
(0.5,0.5,0.5)
Edges:
(0,8)
(1,8)
(2,8)
(3,8)
(4,8)
(5,8)
(6,8)
(7,8)
```

According to the results, we can visualize the scaffold examples as in Figures 13.

## C.2   MORE ANALYSIS FOR PROPERTY PREDICTOR IN AGENT SUPERVISOR

Figure 14 shows the fitting performance of the proposed disentangled predictor on the test set regarding three mechanical properties (*i.e.*, Young's modulus, Shear modulus, and Poisson's ratio). It can be seen that all fitting $R^2 > 0.8$, demonstrating effective fitting performance. Mover, Table 6 compared our proposals with several existing works, including invariant model SphereNet (Liu et al., 2022), Equivariant model (Liao and Smidt, 2023) and ViSNet (Wang et al., 2024), and valina VAE, demonstrating the superiority of disentangled semantic latent in effectively capturing mechanical proeprties.

## C.3   MORE CASE STUDEIS

In this section, we show more case studies given a design concept. Figures 15 and 16 show two other case studies, which demonstrate the effectiveness of multi-agent collaborations. Specifically, Case study 2 proposes a simple scaffold that has extreme flexibility at first, and combines it with a possible structure that contains the most suitable mechanical properties to generate final results. In addition, Case study 3 shows that the evaluation score increases with multi-agent collaboration iterations. Finally, it generates a complicated but suitable structure.

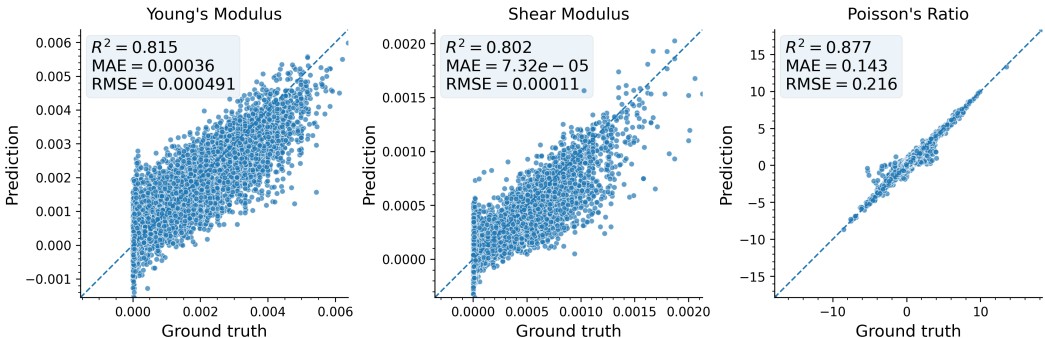

Figure 14: Prediction results of the proposed disentangled VAE on three properties. All $R^2 > 0.8$ demonstrates the strong prediction results on the three mechanical properties.

Table 6: Performance comparison of different models on predicting mechanical properties (MAE). Show the superior performance of the proposed framework.

| Models | Young's Modulus | Shear Modulus | Poisson's Ratio |
|---|---|---|---|
| Vanilla VAE | $6.3e^{-4}$ | $1.4e^{-4}$ | 0.39 |
| SphereNet (Liu et al., 2022) | $4.7e^{-4}$ | $1.0e^{-4}$ | 0.35 |
| Equiformer (Liao and Smidt, 2023) | $6.6e^{-4}$ | $2.2e^{-4}$ | 0.36 |
| ViSNet (Wang et al., 2024) | $6.2e^{-4}$ | $6.3e^{-2}$ | 0.37 |
| MetaSymbO (disentangled VAE) | $\mathbf{3.5e^{-4}}$ | $\mathbf{7.3e^{-5}}$ | **0.14** |

# D    THEORETICAL ANALYSIS

## D.1    LATENT GENERATION MODELS

Auto-encoder (AE) is widely used in the material discovery domain to compress discrete material data to a continuous latent space for downstream tasks (especially inverse material design) (Zeng et al., 2025; Hanakata et al., 2020). VAEs (Kingma and Welling, 2014) introduce a Gaussian prior and derive evidence lower bound optimization (ELBO). More recently, latent diffusion models (LDMs) are extensively explored due to its strong ability in reconstructing high-fidelity data (Rombach et al., 2022; Podell et al., 2023; Fu et al., 2024). Both VAEs and LDMs typically impose a multivariate Gaussian prior on the latent space, thereby enabling the application of symbolic logic operators directly within the latent Gaussian manifold.

Therefore, Agent Generator is instantiated as an AE-based latent generation model in this work, and we implement the basic version of VAEs and LDMs for experiments.

Formally, given a metamaterial $M = (\mathbf{L}, \mathcal{U})$, Both VAEs and LDMs operate by encoding $M$ into a continuous latent variable $\mathbf{z} \in \mathcal{Z} \subseteq \mathbb{R}^d$ via a stochastic encoder: $q_{\phi}(\mathbf{z} \mid M)$, and reconstruct the input through a decoder: $p_{\theta}(M \mid \mathbf{z})$. Similar for VAEs and LDMs, a standard Gaussian prior is imposed over the latent space: $p(\mathbf{z}) = \mathcal{N}(\mathbf{z} \mid \mathbf{0}, \mathbf{I})$. The difference lies in the objectives. Specifically, VAEs conduct ELBO for optimization:

$$\mathcal{L}_{\text{VAE}}(\boldsymbol{\phi}, \boldsymbol{\theta}) = \mathbb{E}_{q_{\boldsymbol{\phi}}(\mathbf{z}\mid M)} \left[\log p_{\boldsymbol{\theta}}(M \mid \mathbf{z})\right] - \text{KL}\left(q_{\boldsymbol{\phi}}(\mathbf{z} \mid M) \,\|\, p(\mathbf{z})\right), \tag{18}$$

while in LDMs, latent variables $\mathbf{z}_0$ are further corrupted over $T$ steps via a diffusion process and denoising process:

$$\begin{aligned} q(\mathbf{z}_t \mid \mathbf{z}_0) &= \mathcal{N}(\mathbf{z}_t \mid \sqrt{\bar{\alpha}_t}\mathbf{z}_0, (1 - \bar{\alpha}_t)\mathbf{I}), \\ p_{\boldsymbol{\theta}}(\mathbf{z}_{t-1} \mid \mathbf{z}_t) &= \mathcal{N}(\mathbf{z}_{t-1} \mid \boldsymbol{\mu}_{\boldsymbol{\theta}}(\mathbf{z}_t, t), \boldsymbol{\Sigma}_{\boldsymbol{\theta}}(t)), \end{aligned} \tag{19}$$

where $\bar{\alpha}_t = \prod_{s=1}^{t} \alpha_s$, and $\{\alpha_t\}$ is a predefined noise schedule. The training objective minimizes the expected noise prediction to learn the parameters $\theta$ in denoising model $\mathcal{L}_{\text{LDM}}(\boldsymbol{\theta}) = \mathbb{E}_{\mathbf{z}_0, \boldsymbol{\epsilon}, t} \|\boldsymbol{\epsilon} - \boldsymbol{\epsilon}_{\boldsymbol{\theta}}(\mathbf{z}_t, t)\|^2$, where $\mathbf{z}_t = \sqrt{\bar{\alpha}_t}\mathbf{z}_0 + \sqrt{1 - \bar{\alpha}_t}\boldsymbol{\epsilon}$, and $\boldsymbol{\epsilon} \sim \mathcal{N}(\mathbf{0}, \mathbf{I})$ is standard Gaussian noise.

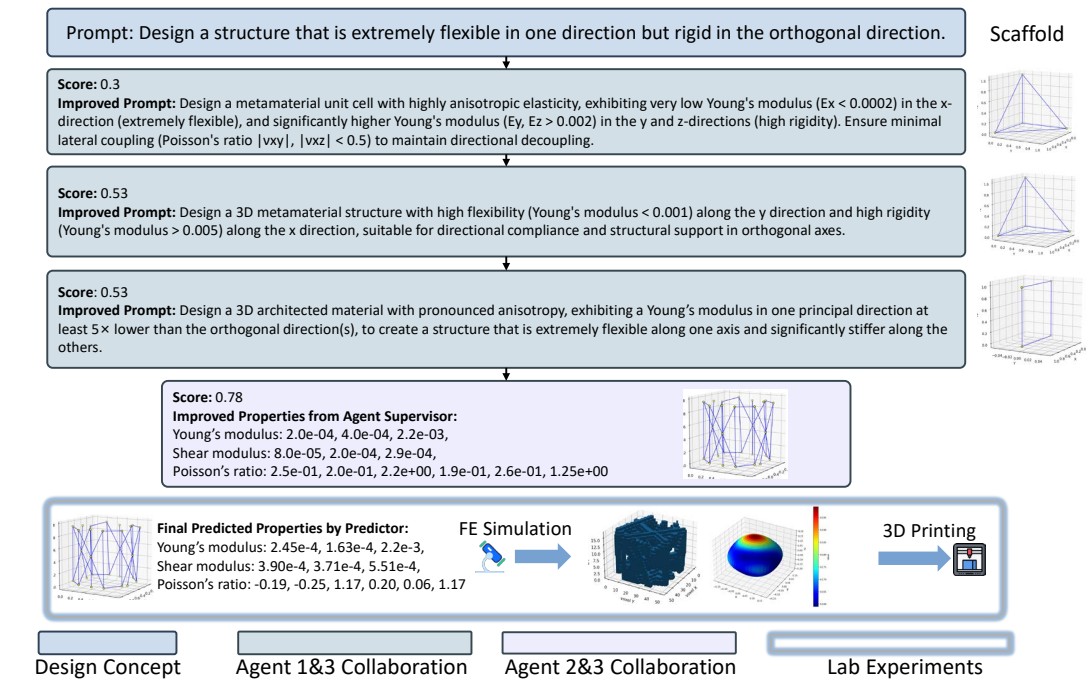

Figure 15: Second case study.

## D.2    DERIVATION OF SYMBOLIC OPERATORS IN GAUSSIAN

### D.2.1    DERIVATION OF MIX OPERATOR (EQ. 10)

Given two diagonal-Gaussian latent posteriors $q_M(\mathbf{z}) = \mathcal{N}(\boldsymbol{\mu}_M, \mathrm{diag}\,\boldsymbol{\sigma}_M^2)$ and $q_{M'}(\mathbf{z}) = \mathcal{N}(\boldsymbol{\mu}_{M'}, \mathrm{diag}\,\boldsymbol{\sigma}_{M'}^2)$, their convex combination is

$$p_{\mathrm{mix}}(\mathbf{z}) = (1 - \lambda_{mix})\,q_M(\mathbf{z}) + \lambda_{mix}\,q_{M'}(\mathbf{z}), \qquad \lambda_{mix} \in [0, 1]. \tag{20}$$

$$\mathbb{E}_{p_{\mathrm{mix}}}[\mathbf{z}] = (1 - \lambda_{mix})\boldsymbol{\mu}_M + \lambda_{mix}\boldsymbol{\mu}_{M'} = \boldsymbol{\mu}_{\mathrm{mix}}.$$

Let $\boldsymbol{\Sigma}_M = \mathrm{diag}\,\boldsymbol{\sigma}_M^2$ and $\boldsymbol{\Sigma}_{M'} = \mathrm{diag}\,\boldsymbol{\sigma}_{M'}^2$. Then

$$\boldsymbol{\Sigma}_{\mathrm{mix}} = (1 - \lambda_{mix})\boldsymbol{\Sigma}_M + \lambda_{mix}\boldsymbol{\Sigma}_{M'} + \lambda_{mix}(1 - \lambda_{mix})(\boldsymbol{\mu}_M - \boldsymbol{\mu}_{M'})(\boldsymbol{\mu}_M - \boldsymbol{\mu}_{M'})^\top. \tag{21}$$

To maintain diagonal structure and avoid the expensive cross term, we drop the outer-product term and interpolate standard deviations:

$$\sigma_{\mathrm{mix},k} \approx (1 - \lambda_{mix})\,\sigma_{M,k} + \lambda_{mix}\,\sigma_{M',k}.$$

With $\boldsymbol{\sigma}_{\mathrm{mix}}$ defined above and a small constant $\varepsilon$ for numerical stability,

$$p_{\mathrm{mix}}(\mathbf{z} \mid \lambda_{mix}) = \mathcal{N}\Big(\mathbf{z};\ (1 - \lambda_{mix})\boldsymbol{\mu}_M + \lambda_{mix}\boldsymbol{\mu}_{M'}, \mathrm{diag}\big(((1 - \lambda_{mix})\boldsymbol{\sigma}_M + \lambda_{mix}\boldsymbol{\sigma}_{M'})^2 + \varepsilon\big)\Big) \tag{22}$$

**Remark.** Equation 22 is a *heuristic single-Gaussian approximation*: it preserves the exact mean but underestimates the true covariance by omitting the cross term in 21. This trade-off yields a numerically stable, fully differentiable latent representation while retaining controllable guidance via $\lambda_{mix}$.

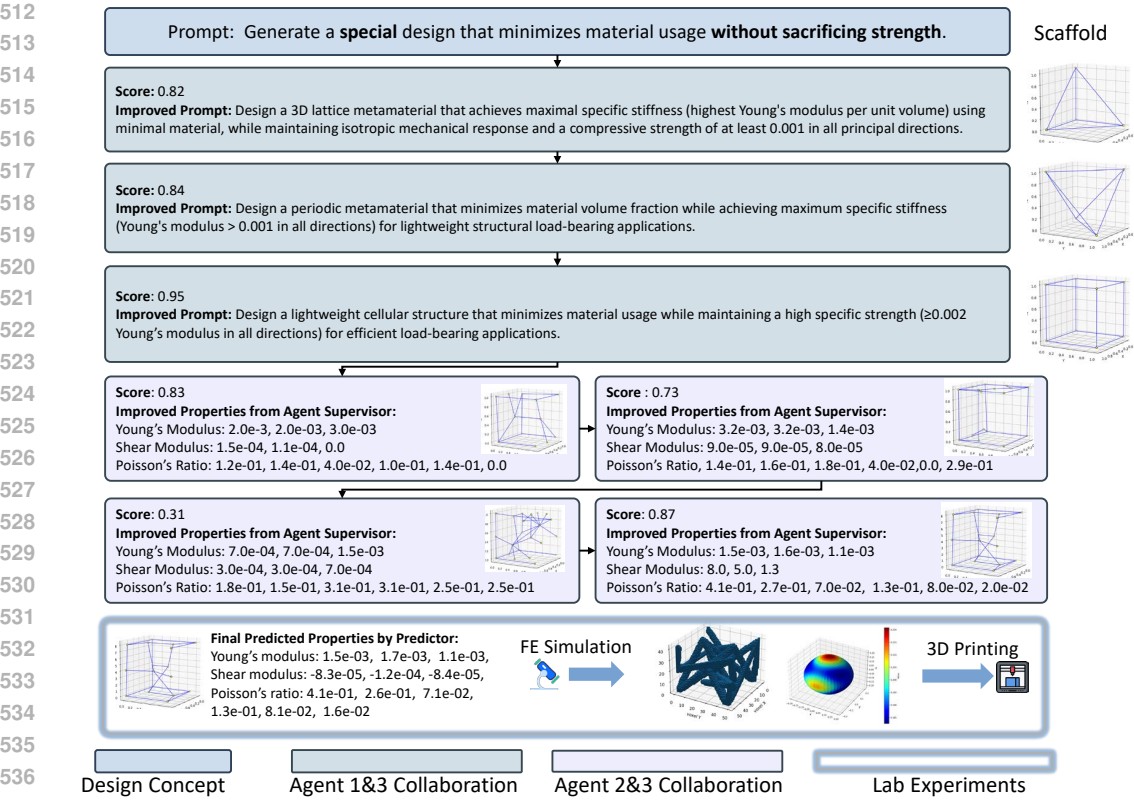

Figure 16: Third case study.

### D.2.2 DERIVATION OF INTERSECTION OPERATOR (EQ. 12)

Recall the formula of PoE (Eq. 11) and the Gaussian distribution of $p_M$ and $p_{M'}$, we have the following derivation by following (Kant et al., 2024; Hinton, 2002b).

$$p_{\text{int}}(\mathbf{z}) \propto p_M(\mathbf{z})\, p_{M'}(\mathbf{z})$$

$$= \underbrace{\frac{1}{(2\pi)^{d/2}|\boldsymbol{\Sigma}_M|^{1/2}} \exp\!\Big[-\tfrac{1}{2}(\mathbf{z}-\boldsymbol{\mu}_M)^{\top}\boldsymbol{\Sigma}_M^{-1}(\mathbf{z}-\boldsymbol{\mu}_M)\Big]}_{p_M(\mathbf{z})}\ \underbrace{\frac{1}{(2\pi)^{d/2}|\boldsymbol{\Sigma}_{M'}|^{1/2}} \exp\!\Big[-\tfrac{1}{2}(\mathbf{z}-\boldsymbol{\mu}_{M'})^{\top}\boldsymbol{\Sigma}_{M'}^{-1}(\mathbf{z}-\boldsymbol{\mu}_{M'})\Big]}_{p_{M'}(\mathbf{z})}$$

$$\propto \exp\!\Big[-\tfrac{1}{2}\Big(\mathbf{z}^{\top}(\boldsymbol{\Sigma}_M^{-1}+\boldsymbol{\Sigma}_{M'}^{-1})\mathbf{z} - 2\mathbf{z}^{\top}(\boldsymbol{\Sigma}_M^{-1}\boldsymbol{\mu}_M+\boldsymbol{\Sigma}_{M'}^{-1}\boldsymbol{\mu}_{M'})\Big)\Big].$$

Let the *precision* matrix be $\boldsymbol{\Lambda}_{\text{int}} = \boldsymbol{\Sigma}_M^{-1} + \boldsymbol{\Sigma}_{M'}^{-1}$, and define $\boldsymbol{\eta}_{\text{int}} = \boldsymbol{\Sigma}_M^{-1}\boldsymbol{\mu}_M + \boldsymbol{\Sigma}_{M'}^{-1}\boldsymbol{\mu}_{M'}$. Completing the square gives

$$p_{\text{int}}(\mathbf{z}) \propto \exp\!\Big[-\tfrac{1}{2}(\mathbf{z}-\boldsymbol{\mu}_{\text{int}})^{\top}\boldsymbol{\Lambda}_{\text{int}}(\mathbf{z}-\boldsymbol{\mu}_{\text{int}})\Big], \qquad \boldsymbol{\mu}_{\text{int}} = \boldsymbol{\Lambda}_{\text{int}}^{-1}\boldsymbol{\eta}_{\text{int}}.$$

Since $\boldsymbol{\Sigma}_{\text{int}} = \boldsymbol{\Lambda}_{\text{int}}^{-1}$, we arrive at

$$p_{\text{int}}(\mathbf{z}) = \mathcal{N}\big(\mathbf{z}\,;\,\boldsymbol{\mu}_{\text{int}}, \boldsymbol{\Sigma}_{\text{int}}\big), \quad \boldsymbol{\Sigma}_{\text{int}} = (\boldsymbol{\Sigma}_M^{-1}+\boldsymbol{\Sigma}_{M'}^{-1})^{-1}, \quad \boldsymbol{\mu}_{\text{int}} = \boldsymbol{\Sigma}_{\text{int}}\big(\boldsymbol{\Sigma}_M^{-1}\boldsymbol{\mu}_M+\boldsymbol{\Sigma}_{M'}^{-1}\boldsymbol{\mu}_{M'}\big).$$

**Remark.** The intersection (PoE) weights each mean by its precision, yielding a sharper Gaussian concentrated where both experts agree.

### D.2.3 DERIVATION OF NEGATION OPERATOR (EQ. 14)

Similar to the derivation of PoE (Kant et al., 2024; Hinton, 2002b), we can derive the following for the negation operator.

Remember that the original and scaffold latent distributions are two multivariate Gaussians

$$p_M(\mathbf{z}) = \mathcal{N}(\boldsymbol{\mu}_M, \boldsymbol{\Sigma}_M), \quad p_{M'}(\mathbf{z}) = \mathcal{N}(\boldsymbol{\mu}_{M'}, \boldsymbol{\Sigma}_{M'}).$$

We suppress the density peaks of $M'$ inside $M$ by defining

$$p_{\text{neg}}(\mathbf{z}) \;\propto\; \frac{p_M(\mathbf{z})^\alpha}{p_{M'}(\mathbf{z})^\beta}, \qquad \alpha, \beta > 0.$$

Because the log-density of any Gaussian is quadratic, we write

$$\log p_{\text{neg}}(\mathbf{z}) = \alpha \log p_M(\mathbf{z}) - \beta \log p_{M'}(\mathbf{z}) + \text{const}$$
$$= -\tfrac{1}{2}\mathbf{z}^\top\big(\alpha\boldsymbol{\Sigma}_M^{-1} - \beta\boldsymbol{\Sigma}_{M'}^{-1}\big)\mathbf{z} + \mathbf{z}^\top\big(\alpha\boldsymbol{\Sigma}_M^{-1}\boldsymbol{\mu}_M - \beta\boldsymbol{\Sigma}_{M'}^{-1}\boldsymbol{\mu}_{M'}\big) + \text{const}.$$

After that, to ensure the Gaussian, we conduct the moment-matched Gaussian approximation. Specifically, treat the quadratic form above as the (unnormalised) log of a new Gaussian. Define the *negation precision*

$$\boldsymbol{\Lambda}_{\text{neg}} \;\triangleq\; \alpha\boldsymbol{\Sigma}_M^{-1} - \beta\boldsymbol{\Sigma}_{M'}^{-1}, \quad (\boldsymbol{\Lambda}_{\text{neg}} \succ 0 \text{ required}),$$

and invert it to obtain the covariance $\boldsymbol{\Sigma}_{\text{neg}} = \boldsymbol{\Lambda}_{\text{neg}}^{-1}$.

Multiplying the linear term by $\boldsymbol{\Sigma}_{\text{neg}}$ gives the mean:

$$\boldsymbol{\mu}_{\text{neg}} = \boldsymbol{\Sigma}_{\text{neg}}\big(\alpha\boldsymbol{\Sigma}_M^{-1}\boldsymbol{\mu}_M - \beta\boldsymbol{\Sigma}_{M'}^{-1}\boldsymbol{\mu}_{M'}\big).$$

Finally, we obtain the approximated Gaussian as

$$p_{\text{neg}}(\mathbf{z}) \;\approx\; \mathcal{N}\big(\mathbf{z};\, \boldsymbol{\mu}_{\text{neg}}, \boldsymbol{\Sigma}_{\text{neg}}\big), \quad \boldsymbol{\Sigma}_{\text{neg}}^{-1} = \alpha\boldsymbol{\Sigma}_M^{-1} - \beta\boldsymbol{\Sigma}_{M'}^{-1}.$$

The approximation is valid only when $\alpha\boldsymbol{\Sigma}_M^{-1} \succ \beta\boldsymbol{\Sigma}_{M'}^{-1}$, ensuring $\boldsymbol{\Lambda}_{\text{neg}}$ (and hence $\boldsymbol{\Sigma}_{\text{neg}}$) is positive definite.

**Remark.** Choosing $\alpha > \beta$ or scaling $\boldsymbol{\Sigma}_{M'}$ slightly upward guarantees the precision matrix remains positive definite, making the negation operator a well-defined Gaussian.

