# OpenReview forum: "LinguaMate: Language‑Guided Metamaterial Discovery via Symbolic-Driven Latent Optimization"
_ICLR.cc/2026/Conference — Submitted to ICLR 2026_

### Official Review · Reviewer_hZHc · 2025-11-01

**Soundness:** 3
**Presentation:** 3
**Contribution:** 2
**Rating:** 4
**Confidence:** 2

**Summary:**

This paper introduces LinguaMate, a multi-agent framework that bridges the gap between natural language prompts and geometric design for metamaterial discovery. It combines an Agent Designer for language space, an Agent Generator for geometric space, and an Agent Supervisor for mechanical property space. LinguaMate outperforms existing generative model baselines and LLM baselines.

**Strengths:**

1. This paper effectively addresses the challenging and significant problem of language-guided metamaterial discovery.

2. The work is clearly written, presenting a methodology that is both intuitive and well-motivated.

**Weaknesses:**

The primary weakness of this paper lies in its choice of baselines for performance comparison. To demonstrate the superiority of LinguaMate, the authors compare it against four material generative models and six standalone LLMs. The comparisons do not include any existing agent-based frameworks that also combine LLMs and generative models.
This paper is not the first to propose an agent-based approach for metamaterial discovery. Its contribution is positioned as a stronger agent framework compared to existing methods like MetaScientist. Therefore, a systematic and direct performance comparison against these established agent-based baselines is essential.
The finding that LinguaMate, a hybrid system, outperforms standalone generative models (which lack language understanding) or isolated LLMs (which lack geometric awareness) is hardly surprising. This comparison does not sufficiently validate the novelty of the proposed method. A rigorous evaluation would require:

1. Direct comparison against state-of-the-art agent baselines (e.g., MetaScientist).

2. An ablation study showing how LinguaMate's novel components improve upon those existing baselines.

**Questions:**

Please refer to the Weaknesses section above.

---

> ### Author Response · Authors · 2025-11-21
> **Response to Reviewer hZHc (1/2)**
>
> Thanks for your time and review. We'll respond to your concerns question-by-question and clarify some misunderstandings.
>
> ### **Positions of this paper.**
> > This paper is not the first to propose an agent-based approach for metamaterial discovery. Its contribution is positioned as a stronger agent framework compared to existing methods like MetaScientist.
>
> We appreciate the reviewer’s perspective, but we would like to clarify that the contribution of this work goes well beyond proposing another multi-agent pipeline. Please allow us to stress our contributions and novelties.
>
> As detailed in the manuscript, the novelty of this work lies in two core advances that have not been explored in prior agent-based metamaterial design systems:
>
> 1. **First integration of multi-modal (language, geometry, property) awareness for metamaterial discovery.**
>
>     Unlike existing multi-agent works which operate almost exclusively in language space, LinguaMate introduces a modality-specialized design framework that jointly reasons over **natural-language design intents (Agent Designer)**, **geometric latent spaces (Agent Generator)**, and **mechanical property reasoning (Agent Supervisor)**.
>
> 2. **A novel symbolic-driven latent optimization mechanism enabling inference-time compositional design.**
>     This work introduces four symbolic logic operators (Union, Mix, Intersection, Negation) defined directly in disentangled Gaussian latent spaces (Sec. 3.1.2). These operators allow (1) semantic fusion between scaffolds and generated structures, (2) controllable structural modification, and (3) **test-time programmatic composition of metamaterials**, something not possible in prior systems such as MetaScientist.
>
>     This mechanism enables LLM agents or non-expert humans to program 3D metamaterial structures at inference time, effectively giving metamaterials “symbolic programmability.” This represents a new capability and goes beyond previous agent frameworks.
>
>
>
> ### **Q1. Direct comparison against state-of-the-art multi-agent baseline MetaScientist.**
> Thanks for your suggestion. We have tried to compare with MetaScientist to construct a comparable LLM multi-agent system. The details are as follows.
> #### **Q1 - Compare with MetaScientist**
> MetaScientist is the only other agentic system that takes text input and outputs a 3D metamaterial structure. However, its scope and technical design are fundamentally different from our proposed method: (1) It focuses on hypothesis generation instead of structure optimization; (2) the synthesis generator is a conditional diffusion model that requires explicit numeric properties, without language semantic awareness.
>
> Importantly, MetaScientist does not provide the code. It provides only a demonstration webpage and does not expose model details, inference interface, or batch generation capabilities. Therefore, **quantitative benchmarking is not feasible**.
>
> However, to still provide a fair comparison, we reproduced three representative prompts in our case studies and directly compared the structures generated by LinguaMate and MetaScientist. **As shown in Table r1**, MetaScientist defaults to well-known topologies (auxetic lattice, Kelvin cell, octet truss), indicating limited exploration ability within the design space. In contrast, LinguaMate which benefits from symbolic-driven latent optimization and multi-modal awareness, produces valid, nontrivial, and novel metamaterials that better reflect prompt semantics. In short, **MetaScientist consistently reproduces classical, existing lattice types, while LinguaMate generates novel, semantically aligned designs.**
>
> **Table r1. Qualitative comparison with MetaScientist.**
> |Prompt|LinguaMate| MetaScientist|
> |-|-|-|
> |Design a metamaterial with a negative Poisson's ratio and high Young's modulus and shear modulus|Complex combination of star-like struts, complex cubic, and 3D stars, as Figure 7 shows |a typical auxetic lattice|
> |Design a structure that is extremely flexible in one direction but rigid in the orthogonal direction.|A complex cubic of 6 faces and some vertical struts with triangle elements, as Figure 15 shown|a typical Kelvin foam|
> |Generate a special design that minimizes material usage without sacrificing strength.| A combination of cubic with only one face, and complex inner structures, as Figure 16 shows |a typical Octet truss|

---

> > ### Author Response · Authors · 2025-11-21
> > **Response to Reviewer hZHc (2/2)**
> >
> > #### **Q1 - Compare with Multi-agent LLM baseline**
> > According to your suggestion, since we do not find any other works that is suitable for comparison, we constructed an LLM-based multi-agent system by referring [r1] and our work. Designer and Supervisor collaboratively refine the design requirements, and then, generator (another LLM) tends to generate according to the refined design prompt. The results are shown in Table r2. It is clear that all LinguaMate variants outperform multi-agent LLM in both validity and diversity. We will publish the code of multi-agent LLMs in our codebase.
> >
> > **Table r2. Comparison between multi-agent LLMs and LinguaMate.**
> > | Versions | $V_S\%\uparrow$ | $V_P\%\uparrow$ | Cov R.% $\uparrow$ | Repeat Ratio% $\downarrow$ |
> > |- |-|-|-|-|
> > | Multi-agent LLM | 71.4 | 9.52 | 41.5% | 51.2 |
> > | LinguaMate (Gemini2.0, Mix) | 64.53 | 91.74 | 93.3 | **0.00** |
> > | LinguaMate (Gemini2.0, Union) | **89.65** | **95.97** | **99.2** | 10.07 |
> > | LinguaMate (GPT4o-mini, Mix) | 76.84 | 94.17 | 98.2 | **0.83** |
> > | LinguaMate (GPT4o-mini, Union) | **91.31** | **98.35** | 98.7 | 7.43 |
> >
> > [r1] Tian, Jie, et al. "A Multi-Agent Framework Integrating Large Language Models and Generative AI for Accelerated Metamaterial Design." arXiv preprint, 2025.
> >
> > ### **Q2. An ablation study showing how LinguaMate's novel components improve upon those existing baselines.**
> >
> >
> > We need to clarify that our experiments have demonstrated the effectiveness of all proposed novel components. To be specific, our ablation studies have shown the effectiveness of each proposed test-time optimization loss term in **Table 3**. **Table 2** has demonstrated the improvements of the proposed LinguaMate to existing LLM baselines and generative model baselines. **Table 2** compares the Union operator and Intersection operators. **Fig. 5** has explicitly shown the effectiveness of each proposed symbolic operator. **Figs. 6 and 14, and Table 6** have shown the superiority of the proposed disentangled generator compared to previous works in the property prediction task.
> >
> > For your convenience, we highlight some results in Tables r2, r3, r4. We believe that these experiments can comprehensively demonstrate the effectiveness and significant improvements to existing works.
> >
> > **Table r3. Comparison with existing LLM and generative SOTA. Showing improvements on existing baselines.**
> > | Approach | $V_S\%\uparrow$ | $V_P\%\uparrow$ | Cov R.% $\uparrow$ | Repeat Ratio% $\downarrow$ | Prompt Guide score (GPT-4.1)$\uparrow$ | Repeat Num.$\downarrow$ |
> > | :--- | :---: | :---: | :---: | :---: | :---: | :---: |
> > | CDVAE | 57.03 | 0.40 | 55.85 | N/A | N/A | N/A |
> > | Deepseek-Reasoning Guo et al. (2025) | 85.5 | 65.3 | 86.9 | 67.7 | 0.4993 | 76 |
> > | LinguaMate (GPT4o-mini, Mix) | 76.84 | 94.17 | 98.2 | **0.83** | 0.5234 | **0** |
> > | LinguaMate (GPT4o-mini, Union) | **91.31** | **98.35** | **98.7** | 7.43 | **0.5531** | 40 |
> >
> >
> > **Table r4: Ablation study. Showing the effectiveness of each proposed latent optimization term.**
> > | Variant | $\mathcal{V}_S \%\uparrow$ | $\mathcal{V}_P \%\uparrow$ | Cov. R%$\uparrow$ |
> > | :--- | :---: | :---: | :---: |
> > | w/o $\mathcal{L}_s$ | 50.9 | 57.1 | 93.1 |
> > | w/o $\mathcal{L}_{p,e}$ | 47.8 | 45.7 | 93.6 |
> > | w/o $\mathcal{L}_r$ | 51.6 | 62.8 | 94.3 |
> > | w/o $\mathcal{L}_{prior}$ | 58.0 | 62.8 | 95.1 |
> >
> > **Table r5: Comparison on predicting mechanical properties (MAE). Showing the superior performance of the disentangled framework.**
> > | Models | Young's Modulus | Shear Modulus | Poisson's Ratio |
> > | :--- | :---: | :---: | :---: |
> > | Vanilla VAE | $6.3e^{-4}$ | $1.4e^{-4}$ | 0.39 |
> > | SphereNet (Liu et al., 2022) | $4.7e^{-4}$ | $1.0e^{-4}$ | 0.35 |
> > | Equiformer (Liao and Smidt, 2023) | $6.6e^{-4}$ | $2.2e^{-4}$ | 0.36 |
> > | ViSNet (Wang et al., 2024) | $6.2e^{-4}$ | $6.3e^{-2}$ | 0.37 |
> > | MetaSymbO (disentangled VAE) | $\mathbf{3.5e^{-4}}$ | $\mathbf{7.3e^{-5}}$ | **0.14** |

---

### Official Review · Reviewer_BkD9 · 2025-11-01

**Soundness:** 3
**Presentation:** 3
**Contribution:** 2
**Rating:** 6
**Confidence:** 3

**Summary:**

This paper introduces LinguaMate, a language-guided metamaterial discovery framework that bridges the gap between natural language reasoning and geometry-aware generative modeling. The method employs a multi-agent system comprising three specialized agents: Designer (LLM-based language interpreter), Generator (geometry synthesizer with disentangled latent spaces), and Supervisor (property predictor and evaluator). The key innovation lies in symbolic-driven latent optimization—a set of four interpretable latent-space operators (Union, Mix, Intersection, Negation)—that enable programmatic semantic composition and cross-modal alignment between language, geometry, and physical properties.

**Strengths:**

- The introduction of programmable operators in Gaussian latent space provides interpretable and compositional control over design semantics, representing a meaningful contribution to controllable generation.
- The paper benchmarks against multiple strong baselines including VAEs, diffusion models, and advanced LLMs (GPT-4o, Gemini 2.0, DeepSeek-Reasoning), demonstrating consistent quantitative and qualitative improvements in validity, diversity, and language alignment.
- The inclusion of case studies involving finite element simulation and 3D printing adds valuable real-world credibility to the proposed system.

**Weaknesses:**

- The paper lacks formal justification for latent logic operators (Union, Mix, Intersection, Negation) semantic preservation properties and guarantees on maintaining manifold validity.
- The ablation study focuses primarily on loss terms but does not isolate the individual contributions of each symbolic operator or the human-in-the-loop component. It remains unclear which operator is most critical for performance, or how much improvement comes from human intervention versus automated agent collaboration.

**Questions:**

- The anonymous repository link appears to be inaccessible. Please verify the link is correctly configured for anonymous access.
- How does the framework handle cases where LLM-generated scaffolds contain geometric inaccuracies or are physically invalid? What mechanisms ensure that the Generator can still produce valid structures when starting from an imperfect scaffold?

---

> ### Author Response · Authors · 2025-11-21
> **Response to Questions (BkD9)**
>
> We appreciate your valuable comments and positive feedback on the novelty and contributions of this work. Follows, we'll respond to your questions.
>
> ### Q1: The anonymous repository link appears to be inaccessible.
>
> Thanks for the reminder. The anonymous repository is commonly used for code access during review, and we verified its accessibility multiple times before the review. After the review period, we also confirmed that the link remains accessible. The temporary issue was likely due to a server-side glitch. The link should now work properly.
>
> ### Q2: How does the framework handle cases where LLM-generated scaffolds contain geometric inaccuracies or are physically invalid? What mechanisms ensure that the Generator can still produce valid structures?
>
>
> **Table r1. Quantitative evaluation on scaffold.**
> ||$V_s$|$V_p$|Prompt Guidance Score|
> |-|-|-|-|
> |Scaffold|40.5|72|0.60|
> |LinguaMate(GPT4o-mini, Union)|91|98|0.55|
>
> Thanks for your question, we'll answer your questions as follows.
>
> **Intuitive Facts**: The scaffold is defined as a substructure (motif) of the complete structure. Therefore, it is not requried to be physically accurate and valid. The main goal is to preserve as more semantic information as possible.
>
> **Observations**: To answer your question, we conduct experiments to test the validity and prompt guidance score of generated scaffold by given our prompt dataset. Table r1 shows that scaffold has **lower validity** while **higher prompt guidance score**, demonstrating the intuitive facts and its ability in semantic preservation.
>
>
> **Mechanisms to improvie validity in generator**: Table r1 also compared the LinguaMate results, showing **significant improvements on validity** and **slightly decrease in prompt guidance score**. To explain, LinguaMate includes four complementary mechanisms that can prevent invalid generations if the scaffold contains inaccuracies:
> 1. **Initialization from guaranteed-valid structures**: During the Generator/Supervisor collaboration, we optionally initialize from a known-valid structure (e.g., octet, cubic, or any dataset sample with verified validity). Both the scaffold and the initialization are encoded into latent space, giving one latent vector in the valid region and another that may lie outside. This injects valid latent information into the optimization.
> 2. **Gaussian latent optimization that avoids low-density (invalid) regions**: The latent space is trained so that valid structures occupy **high-density** regions, while invalid ones fall in **low-density** regions. Instead of decoding directly from the symbolic operator’s closed-form latents, we perform soft, gradient-based optimization toward the scaffold semantics. *This process naturally avoids drifting into low-density invalid regions, keeping the optimized latent within the valid manifold.*
> 3. **Regularization to prevent latent drift**: The latent optimization objective includes an explicit prior term $L_{prior}$ (Eq.(2)) which penalizes deviations from the latent prior, and a regularization term $L_r$ that constrains unmatched nodes/edges. Both terms stabilize optimization and help preserve the validity of the initialization.
> 4. **Superviser final check**: After decoding, Agent Supervisor evaluates both structure validity (symmetry, periodicity) and semantic consistency. Only designs with a score exceeding the threshold $\tau_{G/S}$ are accepted. This provides a final safeguard that filters out any residual invalid generations.

---

> > ### Author Response · Authors · 2025-11-21
> > **Response to Weaknesses (BkD9)**
> >
> > Again, we appreciate your time and the valuable suggestions. However, regarding the two weaknesses, there might be some misunderstandings that we would like to discuss.
> >
> > > The paper lacks formal justification for latent logic operators' semantic preservation properties and guarantees on maintaining manifold validity.
> >
> > **Semantic preservation of latent logic operators.**
> > *Quantitative justification*: Naturally, the prompt-guidance score directly reflects the degree of semantic preservation. As shown in Table 2, all LinguaMate variants achieve higher prompt-guidance scores than standalone LLMs, demonstrating that the symbolic operators help retain semantics during generation. Furthermore, Table r2 shows that the scaffold itself has a high prompt-guidance score; both Union and Mix preserve part of the scaffold semantics while substantially improving validity and diversity.
> >
> > **Maintaining manifold validity.**
> > Manifold validity is ensured by the optimization mechanism (especially through $L_{p,e}$). Our experiments provide justification: (1) LinguaMate achieves up to 98.35% periodicity and 91.31% symmetry, significantly higher than all baselines (e.g., CDVAE Vp = 0.40%, DiffCSP Vs = 34.46%). (2) Ablation results (Table 3) show that removing key manifold-preserving losses drastically reduces validity. For example, removing key $L_{p,e}$ drops periodicity from 94–98% down to 45.7%. (3) Optimization converges **stably** (Table 4), with semantic loss $L_s$ dropping from 28.6 -> 0.78, showing successful semantic transfer from scaffold to output; meanwhile, the geometric terms $L_{p,e}$ coverages from 123 -> 76, indicating the latents stay in high-density valid region, while preserving semantics.
> >
> >
> > > The ablation study focuses primarily on loss terms but does not isolate the individual contributions of each symbolic operator or the human-in-the-loop component. It remains unclear which operator is most critical for performance, or how much improvement comes from human intervention versus automated agent collaboration.
> >
> > **Table r2. Comparisons for Union and Mix operators with different LLM bases.**
> > | Versions | $V_S\%\uparrow$ | $V_P\%\uparrow$ | Cov R.% $\uparrow$ | Repeat Ratio% $\downarrow$ | Prompt Guide score (GPT-4.1)$\uparrow$ | Repeat Num.$\downarrow$ |
> > |- |-|-|-|-|-|-|
> > | LinguaMate (Gemini2.0, Mix) | 64.53 | 91.74 | 93.3 | **0.00** | **0.5464** | **0** |
> > | LinguaMate (Gemini2.0, Union) | **89.65** | **95.97** | **99.2** | 10.07 | 0.4966 | 56 |
> > | LinguaMate (GPT4o-mini, Mix) | 76.84 | 94.17 | 98.2 | **0.83** | 0.5234 | **0** |
> > | LinguaMate (GPT4o-mini, Union) | **91.31** | **98.35** | 98.7 | 7.43 | **0.5531** | 40 |
> >
> >
> > The ablation study is designed to evaluate the importance of each loss term in the symbolic-driven latent optimization process. The role of the symbolic operators is analyzed separately rather than through the ablation table. Their individual contributions are already reported in Table 2 and Fig. 5.
> >
> > To be specific, we compare the contribution of symbolic operators, Union and Mix, as follows with two different LLM baselines in Table r2. The results show that **Union consistently yields higher validity, while Mix provides superior diversity.** In addition, **GPT-4o-mini outperforms Gemini-2.0**, indicating that the capability of LinguaMate can improve with the evolution of stronger LLMs.
> >
> >
> > For Intersection and Negation, these operators are designed for **programmability**, ie, extracting shared semantics or suppressing certain semantic components, rather than serving as end-to-end generators that optimize validity or diversity. Thus, numerical comparison is neither meaningful nor aligned with their design intent. Nevertheless, we qualitatively demonstrate their intended semantic behaviors and structural transformations in Fig. 5, confirming their effectiveness for programmable metamaterial manipulation.

---

### Official Review · Reviewer_6jD7 · 2025-11-02

**Soundness:** 2
**Presentation:** 2
**Contribution:** 2
**Rating:** 2
**Confidence:** 3

**Summary:**

This work introduces a system for the generation of novel metamaterials, microstructures that exhibit particular material properties. These materials are represented by a periodic lattice with nodes and edges forming a repeating structure. The system developed by the authors consists of 3 parts: an "agent designer", which is an LLM that retrieves existing materials from available literature that matches the user's natural-language specified goals, an "agent generator", a VAE that defines a searchable latent space over materials and an "agent supervisor consisting of a ML-based property predictor and LLM for judging the properties of generated materials.  The generator uses an explicitly disentangled decoder structure to facilitate easier generation and control during latent space search along with a set of symbolic operations that allow for guided exploration in the space. By iteratively utilizing these 3 agents the systems generates new candidate Metamaterials. In their experiments, the authors show that their approach generates new valid Metamaterials at a higher rate than prior work.

**Strengths:**

**Motivation**

- The work is well motivated and tackles an under explored area of generative ML for metamaterials design
- The goals of the system and potential use cases are clear. The authors bring together a number of important recent developments in machine learning for this problem including LLM-based prompt understanding and VAEs for periodic structure prediction.

**Novelty**
- I am not an expert in this particular subfield, but I am not aware of prior work that allows for natural language guided design of Metamaterials.
- As far as I know the authors approach to exploring the space of materials is also novel.

**Evaluation**
- The authors compare against a good range of reasonable, recent baselines for this area.
- The authors experimentally validated a generation from the system

Overall, this paper is interesting in its application and direction.

**Weaknesses:**

**Confusing writing**
- It was unclear to me until I looked at the supplement, that the output of the designer was a material specification.
- For example it was very unclear in section 3.1.2 how the symbolic operations to apply are chosen and where the initial material comes from.
- I still don't fully understand the process of generating a material. It would be very helpful to have a walkthrough or pseudo code for the entire process of generating a new material in the main text.

**Missing details**
- Table 1 is not very helpful as many of the symbols are not defined in the main text.
- It's not specified how the likelihoods for the generator are defined as far as I can tell. Details of the architecture, such as number of parameters is also missing.
- Unclear how baselines were adapted for this task. For example, as far as I know CDVAE does not generate edge structures and it's unclear how that was handled in this case

**Methodology**
- The system is very complex, without substantial justification for the complexity. There isn't really a formal justification that the LLM components are improving the performance.
-  Authors claim that the decoder is explicitly disentangled, but the diagram of the implementation suggests there are shared layers in the decoder. I don't understand how to reconcile these two things.
  - Given this, it's unclear how meaningful the symbolic operations actually are.
- The method relies on ML learned evaluation for optimizing for target properties, which could potentially bias the results.

**Evaluation**
- The main goal of this paper is property guided generation, however this isn't properly evaluated. The metrics reported are only validity, diversity and a questionable LLM evaluated metric for language guidance. It seems necessary to compare the properties of materials generated by LinguaMate to baselines by FE simulation or experimentally.
-  The authors claim to have experimentally evaluated a material generated by the model, but give no details about this material.

**Questions:**

- Can you discuss a real-world use case where the natural language goal specification would be important? Wouldn't the applications mentioned in the introduction ("biomedical devices, transportation systems, robotics") generally benefit from engineers specifying exact design requirements?
- Why does the designer not output lattice parameters?
- Do the distributions in the generator optimization come from the encoder?

---

> ### Author Response · Authors · 2025-11-21
> **Response to Reviewer 6jD7 (1/3)**
>
> Thanks for your time for review. We'll address your questions and clarify some misunderstandings in the following.
>
> ## Q1: Discuss a Real-world use case where the natural language goal specification would be important.
>
> First of all, we'd like to clarify that we have explained why natural-language guidance is important in **Lines 58–63**. As stated: "metamaterial discovery often begins with incomplete information, evolving constraints, and only vague conceptual goals", "Natural language, therefore, provides a flexible way to specify qualitative design intents".
>
> By contrast, the assumption that engineers can simply “specify exact requirements” ignores the limitations of traditional methods that literatures have verified. Our cited works in Line 58 clearly stated this. For example, [r1] writes: traditional inverse-design methods 'fail to capture required mechanical behaviors' due to non-uniqueness and the high dimensionality of variables. [r2] writes: "inverse problems are usually intractable,” making explicit numerical targets unreliable or even impossible in early design stages.
>
> Here, we provide a practical use case to explain the limitations: “Design a lattice structure for a robotic fingertip, human-hand sized, with human-like compliance, capable of large repeated deformation, printable on a Formlabs 3D printer.” This real engineering requirement cannot be expressed as a fixed numeric vector. It involves qualitative multi-objective, cross-modal constraints.
>
> Therefore, natural-language guidance is not optional; it is **necessary** for realistic metamaterial design workflows. This question otherwise overlooks our original manuscript, previous literature, and practical engineering reality.
>
> Our work bridges this gap by enabling material design from conceptual, language-level intents, which traditional methods cannot handle.
>
>
> ## Q2: Why does the designer not output lattice parameters?
>
> We need to point out that the multi-agent generation process is a **coarse-to-fine** process where the designer designs a motif-like scaffold, and the generator generates complete materials. In Section 3.1.1, we stated the limitations of LLM's capability in generating complete materials in **Line 199-203**. Further, we explain the design objective of Scaffold in **Line 204-208**. To clarify the question, we repeat the two parts from our manuscript:
>
> 1. (Limitations of LLM Capability) LLM designers are good at language understanding discrete, but they are unable to generate continuous geometries such as lattice parameters. "*The experiments for LLMs in Table 2 show that LLMs tend to generate repeated structures, indicating their capability in successfully retrieving existing structures and limitations in exploration of a large geometric design space. To balance this, we propose “scaffold”.*"
>
> 2. (Design Objectives of Scaffold) "*The high-level idea of it is to utilize the strong retrieving and language-understanding power of LLMs while avoiding its limitations in geometric generation*". The scaffold captures the coarse semantic mechanism (auxetic, chiral) as a reusable motif (only focusing on topology), not a finished periodic material (that requires lattice vectors).
>
>
> ## Q3: Do the distributions in the generator optimization come from the encoder?
> As shown in Figure 3 generator part, all latent distributions are from the disentangled encoder $E_D$. Also, as we explicitly stated in **Line 223-226**: "Specifically, as Agent Generator shown in Figure 3, we first disentangle the latent space to separate Gaussian distributions, and apply the four proposed symbolic logic operators to synthesize the programmed structure under the guidance ofthe  Gaussian latent optimization process."
>
>
> [r1] Ha, Chan Soo, et al. "Rapid inverse design of metamaterials based on prescribed mechanical behavior through machine learning." Nature Communications, 2023.
> [r2] Chen, Wei, et al. "Generative inverse design of metamaterials with functional responses by interpretable learning." Advanced Intelligent Systems, 2025.

---

> > ### Author Response · Authors · 2025-11-21
> > **Response to Reviewer 6jD7 (2/3)**
> >
> > ## W1: Writing clarification
> > > ...the output of the designer is unclear.
> >
> > We would like to clarify that **the output of the Designer (Scaffold) is clearly described multiple times** in the main paper. For example, **line 204** explicitly states "Scaffold, the output of Agent Dsigner..."; **Fig. 3** explicitly shows the output of designer "Scaffold: Nodes:..., Edge:..."; **Line 311** again refers to "a Scaffold from Agent Designer $V_m$". The reviewer’s claim simply overlooks these explicit statements.
> >
> > > How the symbolic operations to apply are chosen.
> >
> > To clarify, "how the symbolic operations are chosen" is not a focus of this paper nor a contribution we claim. The choice of symbolic operator is left intentionally flexible for users. Our experiments already demonstrate the effects, e.g., Fig. 5 visualizes all operators, and Table 2 compares them.
> >
> > > where the initial material comes from.
> >
> > We need to point out that this is **already stated clearly in both Table 1 and Appendix B.2**. Specifically, the initialization is performed via "Init(z|y)," where Init($\cdot$) is instantiated as the 1-nearest neighbor of $y$ by searching the training set.
> >
> > > Cannot understand the entire generating process
> >
> > The full generation workflow is **already given in Fig. 3**. It consists of only two steps:
> > (1) Designer generates scaffold via **Designer\&Supervisor collaboration**;
> > (2) Given scaffold, Generator generates final structure via **Generator\&Supervisor collaboration** until score > $\tau_{G/S}$.
> >
> > ## W2: Missing Details
> > > Table 1...symbols are not defined.
> >
> > We need to point out that this is a misunderstanding. **All symbols in Table 1 are explicitly defined in Lines 311–318**, and **Appendix B.2 provides the full step-by-step description in plain text**. The collaboration mechanism is already thoroughly specified in both the main paper and the appendix.
> >
> > > It's not specified how the likelihoods for the generator are defined. Details of the architecture, such as number of parameters is also missing.
> >
> > The **likelihood formulation is already clearly defined in Eq.(1)** which has explicitly defined the core block of generator includding likelihood, where the reconstruction likelihood term is denoted as $p_{\theta}(M|y,z)$ in Eq.(1).
> >
> > Figure 3 shows the architecture and all modules involved.
> >
> > Thus, both the likelihood and the architecture are described directly in the paper.
> >
> > > Unclear how baselines were adapted for this task ...  CDVAE does not generate edge...
> >
> > The concern is unfounded. Adapting CDVAE and other baselines to our task is trivial: the only modification required is replacing the original dataset with our training and testing data. No architectural change is needed.
> >
> > Furthermore, the claim that “CDVAE does not generate edges,” yet the metrics we applied to CDVAE ( Vs, Vp, Cov R., formally defined in Section B.4.2) evaluate **only node positions**, not edges. These are the very similar metrics used in prior work on CDVAE and related generative models. Therefore, CDVAE is fully compatible with the evaluation protocol.
> >
> > ## W3: Methodology
> > > The system is very complex, without substantial justification for the complexity. There isn't really a formal justification that the LLM components are improving the performance.
> >
> > The justification is explicitly provided by the results. Table 2 clearly shows that combining LLMs (language space) with geometric generative models yields substantial gains over using generative models alone across validity, diversity, and prompt-guidance metrics.
> >
> > > Authors claim that the decoder is explicitly disentangled.
> >
> > We need to point out that this is a misunderstanding. **We never claim the decoder is disentangled**. Throughout the paper we only state that the encoder is disentangled. For example, explicitly, in Fig. 3, we have written "$E_D$ denotes disentangled encoder and $D$ denotes decoder". Fig. 10 also shows disentangling occurs only in the encoder. The reviewer’s comment contradicts what is written.
> >
> > >  how meaningful the symbolic operations actually are.
> >
> > We have already explained and demonstrated their meaning at multiple levels:
> > * Motivation: “motivated by programmable methods… we design a symbolic module to synthesize the language semantics and geometries” (Line 93).
> > * Insight: “combining existing metamaterials can yield novel structures with desired properties; therefore, we propose four symbolic operators” (Line 158).
> > * Design Goal: symbolic operators are introduced “to ensure language guidance and programmability” (Line 243, ).
> > * Quantitative Evidence: Table 2 shows clear and measurable behavioral differences between Mix and Union operators, demonstrating their functional meaning.
> > * Qualitative Evidence: Fig. 5 visualizes how each operator shapes latent semantics, confirming their behavioral interpretability.
> >
> > In short, symbolic operators are not arbitrary. They are motivated, defined, and empirically validated throughout the paper.

---

> > > ### Author Response · Authors · 2025-11-21
> > > **Response to Reviewer 6jD7 (3/3)**
> > >
> > > ## W4: Evaluation
> > > > The main goal of this paper is property-guided generation...
> > >
> > > We need to clarify that this might be a misunderstanding of our work. The main goal is not property-guided generation. The paper’s purpose is **Language-Guided**, **Multi-Modal aware** Metamaterial Discovery, as explicitly stated in **Introduction Line 70**: "Can we have a knowledgeable metmaterial scientist, who has multiple domain experts in geometric topology awareness, flexible natural language understanding and effective metamaterial design."
> > >
> > > Our framework unifies language, geometry, and properties, not merely property conditioning.
> > >
> > > >compare with FE simulation
> > >
> > > We already include FE simulations in the case study. Several generated structures are evaluated with FE to compute stiffness and Young’s modulus, as shown in the colored plots in the bottom of Fig. 7. These FE results are not directly comparable to surrogate predictions because FE uses different modeling assumptions, i.e., boundary conditions, mesh settings, and material normalization, which naturally lead to scale differences.
> > >
> > > More importantly, the paper already provides direct comparisons between predictions and simulation ground truth (GT): **Figs. 6 and 14** compare prediction and GT in three mechanical properties. **Table 6** reports the MAE of predictions and compares with other methods, showing superiority of this work.
> > >
> > > These results demonstrate that our predictor accurately fits simulation-derived mechanical properties, confirming the robustness and reliability of the proposed architecture.

---

### Meta-Review · Area_Chair_cmc7 · 2026-01-07

**Summary:**

The paper proposes a multi-agent, language-guided framework for metamaterial discovery that combines an LLM-based designer, a VAE generator with disentangled latent space, and a property-predicting supervisor.
Reviewers agree the problem is important and the idea is intuitively appealing, but they converge on three common weaknesses:
(a) Baseline selection: comparisons are limited to standalone LLMs or pure generative models; no direct quantitative comparison with prior agent-based systems (e.g., MetaScientist).
(b) Ablation depth: individual contribution of each symbolic operator and the human-in-the-loop component is not isolated.
(c) Property evaluation: although FE simulation is shown, no systematic head-to-head comparison of properties (stiffness, Poisson ratio, etc.) between LinguaMate and baseline-generated designs is provided; metrics focus on validity, diversity, and an LLM-scored “prompt-guidance” measure.

**Reviewer Concerns:**

Addressed:
Clarity of scaffold output, initialization strategy, likelihood definition, and repository accessibility.
Added ablation of symbolic operators (Union vs. Mix) and qualitative comparison with MetaScientist.
Outstanding:
No quantitative property-matching comparison versus prior agent frameworks.
No per-operator ablation in the main validity/diversity table.
No human-vs-automatic ablation to quantify designer supervision gain.

**Reviewer Scores:**

6JD7 (R1): 2 → 2
BkD9 (R2): 6 → 6
hZHc (R3): 4 → 5

---

### Decision · Program_Chairs · 2026-01-26

Reject